# Biallelic variants in the noncoding RNA gene *RNU4-2* cause a recessive neurodevelopmental syndrome with distinct white matter changes

Genetic variants in *RNU4-2*, which is transcribed into the U4 small nuclear RNA component of the major spliceosome, were recently shown to cause ReNU syndrome, a prevalent dominant neurodevelopmental disorder (NDD). These variants almost exclusively arise de novo and cluster within 18 nucleotides of *RNU4-2*. Here we describe a new recessive NDD associated with homozygous and compound heterozygous variants in *RNU4-2*. We identify 38 individuals with biallelic variants outside the 18-nucleotide ReNU syndrome region that cluster within other functionally important elements of U4: Stem II, the k-turn and the Sm protein binding site. We characterize the clinical phenotype in 31 individuals, demonstrating that the recessive disorder is clinically distinct from ReNU syndrome and is associated with distinctive white matter abnormalities, including enlarged perivascular spaces. Finally, we find reduced *RNU4-2* transcript levels in individuals with the recessive disorder, suggesting a loss-of-function disease mechanism that is distinct from the mechanism underlying ReNU syndrome. Together, these findings expand the genotypic and phenotypic spectrum of *RNU4-2*-associated NDDs.

Splicing is a core cellular process mediated by a large ribonucleoprotein (RNP) complex called the spliceosome, which consists of small nuclear RNAs (snRNAs) and many associated proteins[1]. The major spliceosome is responsible for catalyzing the removal of ~99.5% of introns through U2-dependent splicing, while the minor spliceosome is responsible for the remaining 0.5%, which are U12-type introns[2]. Genetic variants in many components of both the major and minor spliceosomes cause a heterogenous group of disorders that are collectively termed 'spliceosomopathies'[3].

While most spliceosomopathies discovered to date are caused by variants in the protein components of the spliceosome, an increasing number involve snRNAs. The first to be identified were recessive disorders caused by variants in the minor spliceosome snRNAs *RNU4ATAC* (HGNC: 34016)[4] and *RNU12* (HGNC: 19380)[5,6]. More recently, de novo variants in *RNU4-2* (HGNC: 10193), which is transcribed into

the U4 snRNA of the major spliceosome, were shown to cause ReNU syndrome, a dominant syndromic neurodevelopmental disorder (NDD)[7,8]. Subsequent to the discovery of ReNU syndrome, variants in two additional major spliceosomal snRNAs, *RNU2-2* (HGNC: 10152) and *RNU5B-1* (HGNC: 10212), were also identified to cause dominant NDDs[9–11]. Furthermore, variants in *RNU4-2* and U6 encoding genes were shown to cause dominant isolated retinitis pigmentosa[12]. Across all of the spliceosomopathies, ReNU syndrome is the most prevalent, predicted to correspond to ~0.4% of all severe NDDs, or around 100,000 individuals worldwide[7].

In a companion paper[13], we describe a saturation genome editing (SGE) experiment, where we simultaneously measured the functional impact of variants across *RNU4-2* in a haploid cell line. These data led us to identify a new recessive NDD caused by biallelic variants outside the T-loop and Stem III regions in which heterozygous variants cause

✉e-mail: cas.simons@populationgenomics.org.au; nwhiffin@well.ox.ac.uk

ReNU syndrome. These biallelic variants map to regions of U4 that are important for binding to other spliceosome components: the Stem II region of interaction with U6, the Sm protein binding site that is essential for snRNA biogenesis and stability, and the k-turn structure in the 5′ stem loop that binds to SNU13/15.5k. Strikingly, variants in the equivalent regions and nucleotides of U4ATAC (the minor spliceosome homolog of U4) cause the RNU4atac-opathies microcephalic osteodysplastic primordial dwarfism type I (MOPDI) or Taybi–Linder syndrome (OMIM: 210710), Lowry–Wood syndrome (OMIM: 226960) and Roifman syndrome (OMIM: 616651)[13,14].

In this study, we describe the clinical phenotype of this new autosomal recessive NDD caused by variants in *RNU4-2*. We identify a cohort of 38 individuals with biallelic variants in specific regions of *RNU4-2* and characterize the clinical phenotype of the recessive NDD syndrome in a subset of 31 individuals. We show that, while this recessive disorder has phenotypic similarities with dominant ReNU syndrome, some features, including specific white matter changes, are distinct. These findings expand the phenotypic and genotypic spectrum of NDDs associated with *RNU4-2*. Furthermore, we establish recessive disease and variant features critical for facilitating accurate diagnosis.

## Results

### Biallelic variants in *RNU4-2* are enriched in individuals with NDD

Our companion manuscript[13] describes an SGE experiment where we measured the functional impact of variants across *RNU4-2*. Each variant ($n = 539$) was given a 'function score' as a measure of the depletion of cells with the variant across two time points. Variants with a function score of less than −0.302 were determined to be significantly depleted and hence to have an effect on cell viability through affecting *RNU4-2* function. Initially, we searched rare disease cohorts for undiagnosed individuals with biallelic variants with significant SGE function scores in regions of *RNU4-2* not yet associated with NDD[13]. This resulted in the identification of 20 individuals from 13 families: ten individuals (including three pairs of siblings) with homozygous variants and ten individuals (including four pairs of siblings) with compound heterozygous variants. We also identified one individual in Genomics England who was classified as diagnosed, but who was compound heterozygous for two *RNU4-2* variants with significant SGE function scores (individual 17; see below).

In this study, we expanded our analysis to search for additional undiagnosed families with NDD with biallelic variants in *RNU4-2* in global rare disease cohorts (Methods), including those with nonsignificant SGE scores. We identified 43 individuals across 33 families, including the 20 reported in ref. 13 (Supplementary Table 1). Fifteen individuals from 11 families had homozygous variants, and 27 individuals from 22 families had compound heterozygous variants. Eight individuals from five families, all of whom had homozygous variants, had consanguineous parents. In the Genomics England 100,000 Genomes Project, we identified biallelic *RNU4-2* variants in six of 5,386 trios with undiagnosed NDD, compared to zero of 4,776 trios with non-NDD phenotypes (one-sided Fisher's exact test $P = 3.3 \times 10^{-3}$).

We identified only 11 individuals in the UK Biobank (UKB) with biallelic variants in *RNU4-2*. Five of the 490,541 genome-sequenced participants had homozygous variants and six individuals from a subset of 200,011 participants with phased genome sequencing data had compound heterozygous variants (Supplementary Table 2). Of the 14 unique variants observed in the UKB individuals in the homozygous or compound heterozygous state, only one (observed as compound heterozygous with a neutral scoring variant) had a significant SGE score (n.120T>C, SGE = −1.15). None of the 11 individuals had any evidence of neurodevelopmental or severe neurological phenotypes. All nine with information on the age at which they left education attended up to at least age 15. Four had a degree (4 of 11; 36.4% versus 47.7% across the full cohort), ten of 11 were reported as 'able to work'

(90.9% versus 93.4% across the full cohort) and none were outliers for fluid intelligence scores.

For each individual in the UKB and cohort with NDD, we calculated the mean SGE function score for the variants identified on their two alleles. Individuals with biallelic variants in the UKB had significantly weaker mean SGE scores than individuals with NDD (UKB mean = −0.076; NDD mean = −0.525; two-sided Mann–Whitney *U*-test $P = 2.1 \times 10^{-5}$; Fig. 1). We excluded five individuals with NDD with mean SGE scores similar to those observed in the general population from further characterization, using a threshold of −0.15 that maximally separated individuals in the UKB from those with NDD (Fig. 1a). This threshold provides a distinction for analysis but should be interpreted as a pragmatic case definition for this initial characterization rather than a definitive threshold (see Discussion). This led to a cohort of 38 individuals from 28 families that we used for all following analyses (Supplementary Table 1 and Extended Data Fig. 1).

### Clinical characterization of the biallelic *RNU4-2*-associated NDD
Of the 38 included individuals with biallelic *RNU4-2* variants, detailed clinical data were available for 31 (21 males, 10 females) through contact with their clinical teams (Fig. 2a, Table 1 and Supplementary Table 3). The median age at the last follow-up was 10 years (range: 6 weeks to 32 years). Most individuals had infantile-onset phenotypes ($n = 16$; 51.6%), while ten (32.2%) had congenital or neonatal onset and five (16.1%) had childhood onset.

Global developmental delay (GDD) or intellectual disability (ID) was present in all individuals ($n = 30$). Severity data, available for 29 individuals, revealed severe GDD in 15 (51.7%) and moderate GDD in 13 (44.8%). All individuals ($n = 29$; 100%) exhibited delayed language development. Fifteen (51.7%) individuals older than 2 years were nonverbal or had no meaningful spoken language. Of the 13 individuals with some expressive language, all reported delayed first words (median = 4 years, range = 20 months to 8 years) and a limited expressive output, ranging from fewer than ten words to simple sentences. Inability to walk was reported in 7 of 25 individuals older than 5 years (28.0%). Among the 18 individuals who achieved ambulation, first steps were consistently delayed, with a median of 2.5 years ranging from 1.4 years to over 5 years. Behavioral abnormalities were reported in 14 of 23 (60.9%), including obsessive–compulsive traits ($n = 6$), aggression or self-injurious behavior ($n = 6$), and emotional lability, tantrums or meltdowns ($n = 6$).

Most individuals had a history of hypotonia (83.9%; 26 of 31), with neonatal onset in 64% (19 of 26), which was commonly associated with feeding difficulties. Spasticity was present in 15 of 30 individuals (50.0%), while movement and coordination abnormalities were observed in 13 of 30 individuals (43.3%), nine with ataxia, three with dystonia, one with choreoathetosis and one unspecified abnormality of coordination. Seizures occurred in 19 of 31 individuals (63.3%) with a median onset of 2.2 years. Seizure types at onset varied and included tonic clonic, atonic, focal, generalized, febrile, absence and startle-triggered seizures. In most cases, seizure semiology evolved over time. Among the 19 individuals with seizures, 17 received antiseizure treatment. Most were treatment-responsive: 13 received monotherapy and four required more than one antiseizure medication. Three individuals (15.8%) had pharmacoresistant epilepsy with persistent daily seizures. No individuals experienced status epilepticus (Supplementary Table 3).

Genital anomalies were observed in 9 of 21 males (42.8%), including micropenis, cryptorchidism, hypoplastic scrotum and testicular ectopia. No genital abnormalities were reported in females ($n = 10$). Heterogenous integumentary abnormalities were noted in 17 of 28 individuals (60.7%), with features such as hypertrichosis ($n = 3$), livedo reticularis ($n = 2$), acrokeratosis verruciformis of Hopf ($n = 2$), hypoplastic nails ($n = 2$), eczema ($n = 2$), hypopigmented macular lesions ($n = 2$), striae distensae ($n = 1$) and pigmentary changes ($n = 1$).

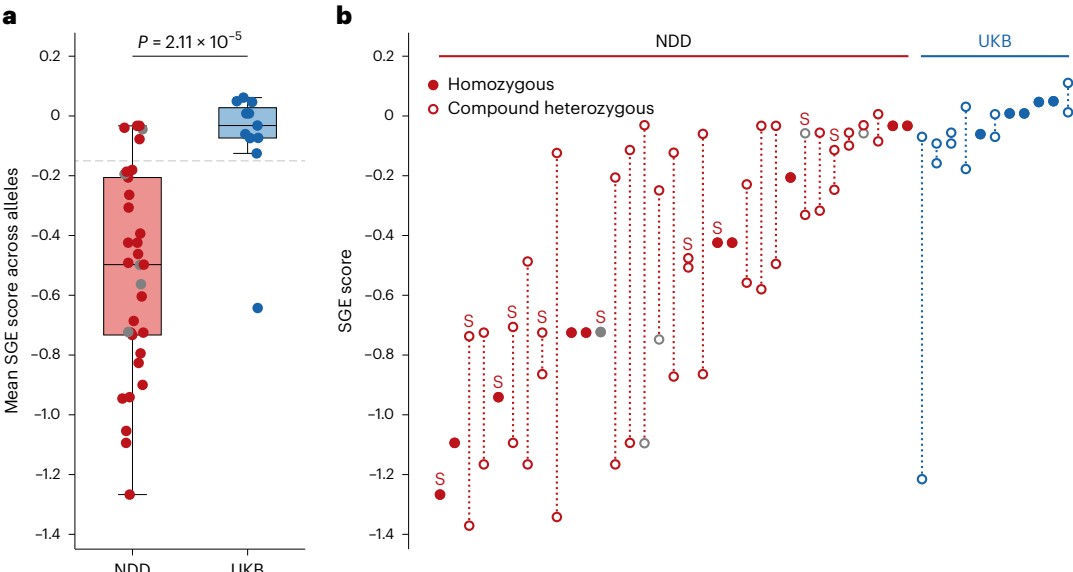

**Fig. 1 | SGE function scores for individuals with biallelic *RNU4-2* variants with NDD versus individuals in the UKB. a**, Mean SGE scores across both alleles per individual. Only one individual from each sibling pair was included. The box and whisker plots show the median and quartiles ± 1.5 times the interquartile range (IQR) of the data. Mean scores were compared with a two-sided Mann–Whitney *U*-test testing 33 cases with NDD against 11 UKB controls. A mean SGE score of less than −0.15 (gray dashed line) was used to include participants in the characterized cohort with NDD. **b**, SGE scores per allele for each individual. Variants from sibling pairs are annotated with the letter 'S'. Across both **a** and **b**, for insertions and deletions without available SGE data, the SGE score was inferred from the mean SGE score across all single-nucleotide variants (SNVs) within the deleted nucleotides or taking the mean SGE score across all SNVs within the nucleotides directly flanking the insertion (Methods and Extended Data Fig. 4). Individuals (**a**) and variants (**b**) with these inferred scores are shown in gray.

There was phenotypic concordance across the seven multiplex families with detailed clinical data available (14 affected individuals). Siblings consistently shared a similar phenotype, including age at onset, systems affected and a comparable degree of ID (Supplementary Table 3). In three families (F2, F4, F5), we observed variable expressivity: in each sibship, one sibling achieved independent walking whereas the other remained nonambulant. In family F2, individual 2 had more marked dilation of perivascular spaces than his sibling, developed spasticity and experienced seizure onset approximately 10 years later.

Individual 17 was classified as diagnosed by Genomics England with a likely pathogenic variant in *GLI3* (NM_000168.6:c.804_810del, heterozygous). However, this *GLI3* variant did not explain all of their reported phenotypes, including white matter abnormalities and microcephaly. The absence of polydactyly in this individual was also inconsistent with *GLI3*-related disorder (OMIM IDs: 175700 and 146510).

**Neuroimaging reveals consistent white matter involvement**

Brain magnetic resonance imaging (MRI) data or reports were available for 27 individuals. Among these, 24 individuals showed abnormalities, most commonly including dilation of perivascular spaces in the periventricular white matter and white matter volume loss (Supplementary Table 4). Only three individuals with imaging performed after 1 year of age were reported to have normal MRIs (individuals 18, 32 and 34), although these images were not available for review.

For 13 individuals, imaging data were directly reviewed by the same pediatric neuroradiologist, enabling detailed comparison (Fig. 3 and Supplementary Table 4). Cerebral white matter changes were the most consistent feature, present in 100% (13 of 13), most frequently manifesting as dilation of perivascular spaces in the periventricular and deep white matter. In more severe cases, this pattern resembled tightly packed microcysts (*n* = 6; individuals 2, 3, 5, 9, 27 and 31). Two individuals whose MRI was performed at less than 2 years old showed only minimal dilation (individuals 4 and 21). Two individuals (2 and 3) underwent serial imaging from infancy: their initial scans showed only

ventriculomegaly, but follow-up studies revealed progressive pathology, including perivascular space dilation, white matter volume loss and cerebellar atrophy (Fig. 3; aged 3 years and 2 years, respectively). While individuals with the most severe dilation of perivascular spaces had lower mean SGE scores across their two variant alleles than individuals with minimal dilation evident on MRI review, this difference was not statistically significant (siblings excluded; *n* = 5 per group; −0.753 (s.d. = 0.389) versus −0.339 (s.d. = 0.233); *P* = 0.056; two-sided Mann–Whitney *U*-test).

**Dominant and recessive *RNU4-2* NDDs have distinct phenotypic features**

We compared the phenotypes observed in 31 individuals with the recessive disorder to 178 individuals from two large nonoverlapping studies of dominant ReNU syndrome (49 from ref. 7 and 129 from ref. 11). Many phenotypes are observed at similar frequencies across the recessive and dominant *RNU4-2* disorders, including GDD (100% and 99.4%, respectively), marked speech and language delay (100% and 92.7%) and seizures (61.3% and 64.4%) (Fig. 2b and Supplementary Table 5). Beyond neurodevelopmental phenotypes, the eye (77.4% and 62.5%) and skeletal system (30.0% and 43.8%) are commonly affected in both disorders.

However, the recessive disorder presents with some distinct phenotypic features that allow clinical differentiation. Whereas non-specific white matter changes are common in ReNU syndrome (23 of 46; 50.0% in ref. 7), the recessive disorder frequently presents dilated perivascular spaces that can mimic a compact microcystic appearance (18 of 23; 78.3%; Fig. 3 and Supplementary Table 4) that had not been reported in ReNU syndrome. Cerebellar atrophy is observed in 50.0% (12 of 24) of the recessive individuals showing a significant enrichment in the recessive disorder (odds ratio = 22.0; 95% CI = 4.32–112.0; false discovery rate (FDR)-corrected *P* = 4.1 × 10⁻⁴; two-sided Fisher's exact test; Fig. 3b). Cerebellar atrophy was not limited to a specific portion of the cerebellum and both cerebellar hemispheres and vermis could be involved. To confirm these reported MRI differences, the same

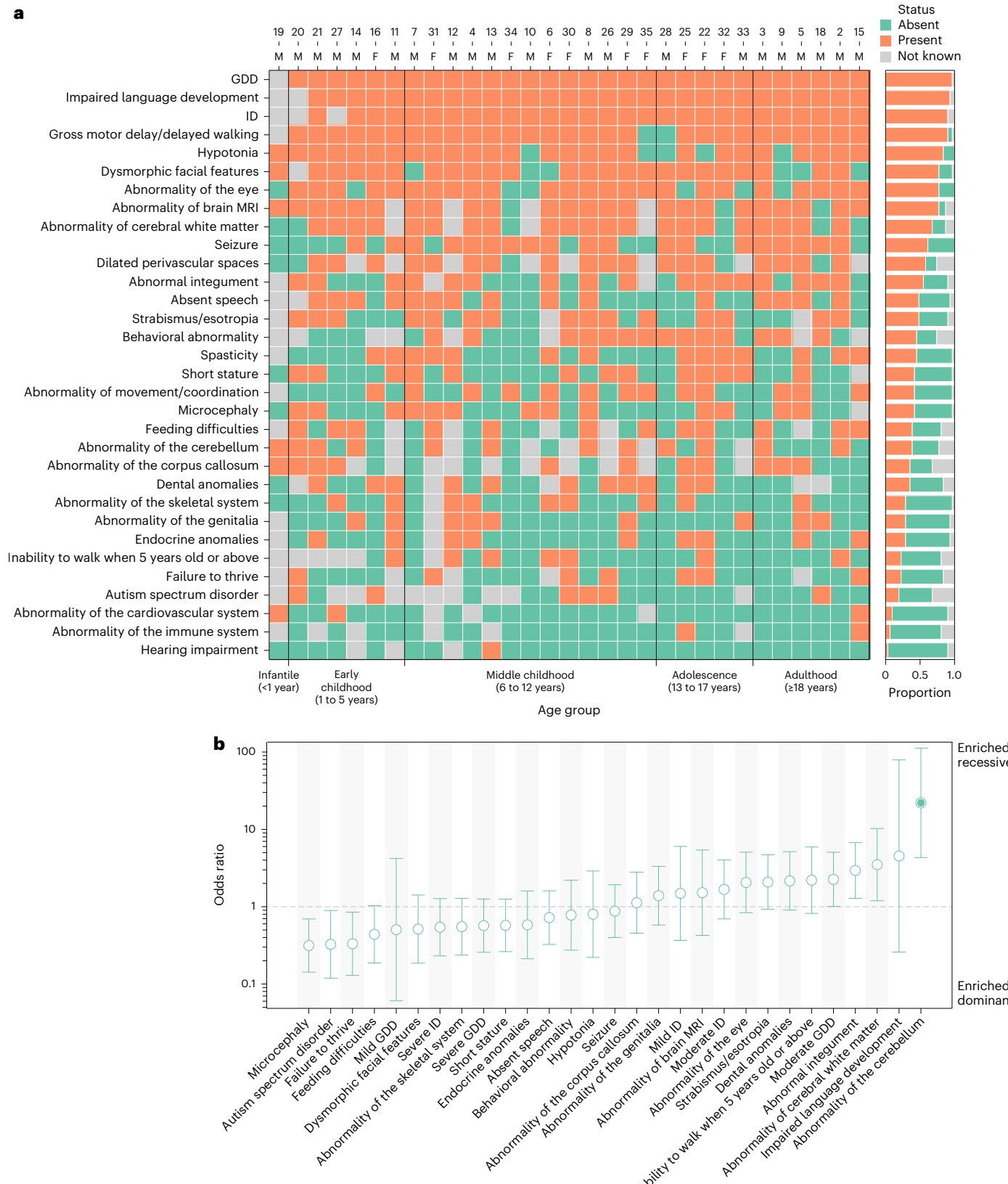

**Fig. 2 | Clinical features of individuals with biallelic *RNU4-2* variants.**
**a**, Color map illustrating Human Phenotype Ontology (HPO) terms present in 31 individuals with the recessive *RNU4-2* NDD and detailed clinical data available. Individuals are stratified according to age at the last evaluation. The sex and individual number (used throughout the text and in Supplementary Tables 1 and 3) are shown across the top. **b**, Phenotype enrichment or depletion in 31 individuals with the recessive *RNU4-2* NDD versus 178 individuals with dominant

ReNU syndrome. Phenotypes are limited to those observed in at least 25% of individuals in this study or the combined ReNU cohorts. The circles mark the odds ratios, with the filled circles showing significantly enriched phenotypes after two-sided Fisher's exact tests with FDR (Benjamini–Hochberg) correction. Odds ratios were calculated with the Haldane–Anscombe correction. The error bars show the 95% confidence intervals (CIs).

**Table 1 | Summary of phenotypes observed in 31 individuals with biallelic *RNU4-2* variants**

| | Both alleles P/LP % (present/known) | Other % (present/known) | Total % (present/known) |
|---|---|---|---|
| Age last seen (median, range) | 14 years, 0.12–32 | 10 years, 2–21 | 10 years, 0.12–32 |
| Male | 100% (8/8) | 56.6% (13/23) | 67.8% (21/31) |
| Female | 0% (0/8) | 43.4% (10/23) | 32.3% (10/31) |
| GDD | 100.0% (7/7) | 100.0% (23/23) | 100.0% (30/30) |
| Severe | 42.9% (3/7) | 54.5% (12/22) | 51.7% (15/29) |
| Moderate | 57.1% (4/7) | 40.9% (9/22) | 44.8% (13/29) |
| ID | 100.0% (6/6) | 100.0% (23/23) | 100.0% (29/29) |
| Severe | 40.0% (2/5) | 52.4% (11/21) | 50.0% (13/26) |
| Moderate | 60.0% (3/5) | 57.1% (12/21) | 57.7% (15/26) |
| Mild | 0% (0/5) | 14.3% (3/21) | 11.5% (3/26) |
| Impaired language development | 100.0% (6/6) | 100.0% (23/23) | 100.0% (29/29) |
| Absent speech | 50.0% (3/6) | 52.2% (12/23) | 51.7% (15/29) |
| Gross motor delay/delayed walking | 100.0% (7/7) | 91.3% (21/23) | 93.3% (28/30) |
| Inability to walk when 5 years old or above | 16.7% (1/6) | 31.6% (6/19) | 28.0% (7/25) |
| Abnormality of brain MRI | 100.0% (7/7) | 85.0% (17/20) | 88.9% (24/27) |
| Abnormality of cerebral white matter | 57.1% (4/7) | 85.0% (17/20) | 77.8% (21/27) |
| Dilated perivascular spaces | 66.7% (4/6) | 82.4% (14/17) | 78.3% (18/23) |
| Abnormality of the corpus callosum | 57.1% (4/7) | 50.0% (7/14) | 52.4% (11/21) |
| Abnormality of the cerebellum | 57.1% (4/7) | 47.1% (8/17) | 50.0% (12/24) |
| Hypotonia | 75.0% (6/8) | 87.0% (20/23) | 83.9% (26/31) |
| Dysmorphic facial features | 57.1% (4/7) | 87.0% (20/23) | 80.0% (24/30) |
| Abnormality of the eye | 62.5% (5/8) | 82.6% (19/23) | 77.4% (24/31) |
| Strabismus/esotropia | 42.9% (3/7) | 57.1% (12/21) | 53.6% (15/28) |
| Seizure | 62.5% (5/8) | 60.9% (14/23) | 61.3% (19/31) |
| Behavioral abnormality | 60.0% (3/5) | 61.1% (11/18) | 60.9% (14/23) |
| Spasticity | 28.6% (2/7) | 52.2% (12/23) | 46.7% (14/30) |
| Abnormal integument | 57.1% (4/7) | 59.1% (13/22) | 60.7% (17/28) |
| Microcephaly | 42.9% (3/7) | 43.5% (10/23) | 43.3% (13/30) |
| Feeding difficulties | 57.1% (4/7) | 44.4% (8/18) | 48.0% (12/25) |
| Short stature | 14.3% (1/7) | 52.2% (12/23) | 43.3% (13/30) |
| Abnormality of movement/coordination | 42.9% (3/7) | 43.5% (10/23) | 43.3% (13/30) |
| Dental anomalies | 0% (0/7) | 57.9% (11/19) | 42.3% (11/26) |
| Abnormality of the genitalia | 14.3% (1/7) | 36.4% (8/22) | 31.0% (9/29) |
| Abnormality of the skeletal system | 12.5% (1/8) | 36.4% (8/22) | 30.0% (9/30) |
| Endocrine anomalies | 28.6% (2/7) | 31.8% (7/22) | 31.0% (9/29) |
| Autism spectrum disorder | 14.3% (1/7) | 35.7% (5/14) | 28.6% (6/21) |
| Failure to thrive | 28.6% (2/7) | 26.3% (5/19) | 26.9% (7/26) |

Only phenotypes observed in at least 20% of individuals are included. Separate counts are shown for (1) individuals with biallelic P/LP variants, (2) individuals where one or both alleles had VUS and (3) all individuals.

pediatric neuroradiologist who reviewed the images from individuals with the recessive NDD also reviewed the MRI images for nine individuals with ReNU syndrome (Supplementary Table 6). While three of nine individuals with ReNU syndrome had mildly dilated perivascular spaces in the periventricular region, none had the severe dilation mimicking a confluent microcystic appearance that is characteristic of the recessive disorder. Furthermore, none of the nine individuals with ReNU had cerebellar atrophy. Thinning of the corpus callosum was a common feature in both ReNU syndrome and the recessive NDD.

The prevalence of dysmorphic facial features was not significantly different between the two disorders (odds ratio = 0.51; 95%

CI = 0.19-1.42, FDR $P$ = 0.34); however, the reported dysmorphic features differ. Dominant ReNU syndrome usually features a myopathic facial appearance with deep-set eyes, epicanthal folds, a broad nasal bridge and anteverted nares, large cupped ears, full cheeks, a tented philtrum and a triangular open mouth with full lips, downturned corners and an everted lower lip vermilion[11]. In contrast, the dysmorphic features in the recessive condition were variable, including a high anterior hairline, synophrys, strabismus, upslanted palpebral fissures, broad nasal bridge and base, bulbous tip, a thin upper lip and dental diastema (Fig. 4a). To further investigate the distinction in facial features, we used the GestaltMatcher framework[15] to compare facial

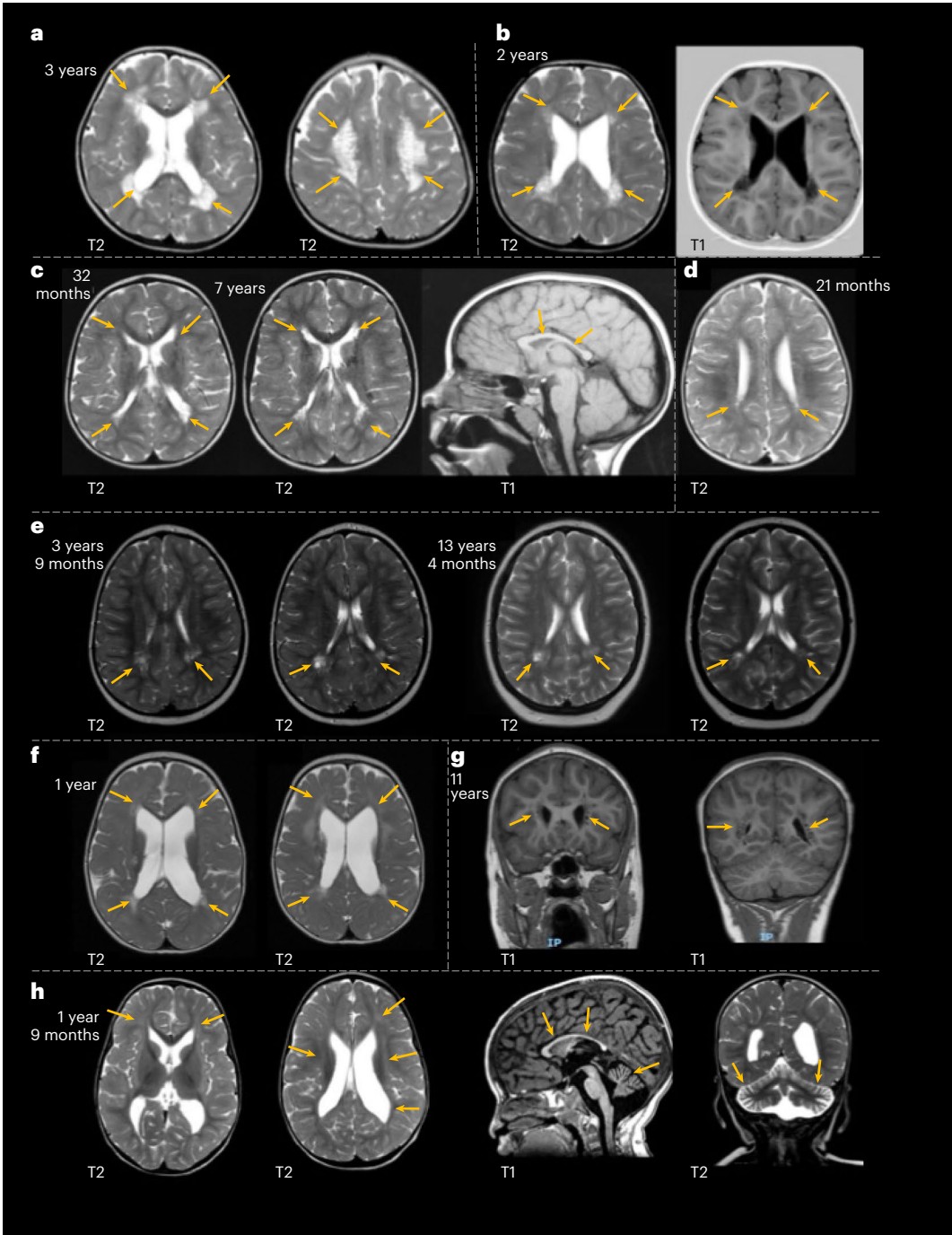

**Fig. 3 | Individuals with biallelic *RNU4-2* variants have consistent cerebral white matter abnormalities.** The yellow arrows point out features of interest in each image. **a,b,** Axial MRI images of individuals 2 (**a**) and 3 (**b**) show extensive dilated perivascular spaces in the periventricular region mimicking a tightly packed microcystic pattern and white matter volume loss. **c,** In individual 25, the extent of dilated perivascular spaces slightly increased between the scans at 32 months and 7 years, with T2-weighted axial views showing progression and a T1-weighted sagittal view at 7 years showing a thin corpus callosum. **d,** T2-weighted axial view of individual 26, the sibling of individual 25, at 21 months shows less severe dilation of the periventricular perivascular spaces. **e,** T2-weighted axial views of individual 28 at ages 3 and 13 years show periventricular focally dilated perivascular spaces in the peritrigonal region bilaterally. **f,** T2-weighted axial view of individual 31 at 1 year shows periventricular dilated perivascular spaces, other patchy areas of white matter signal abnormality, low volume of the white matter and ventriculomegaly. **g,** T1-weighted coronal view of individual 5 shows periventricular dilated perivascular spaces at 11 years. **h,** The MRI scan of individual 21 (axial and coronal T2-weighted and sagittal T1-weighted images) at 1 year shows faint T2 hyperintensities in the white matter, low white matter volume and ventriculomegaly along with a thin corpus callosum and atrophy of the cerebellum.

photographs for 90 individuals with ReNU syndrome, 11 individuals with biallelic variants in *RNU4-2* and 100 'random' individuals with different disorders. We calculated the distance between each pair of faces in clinical face phenotype space (CFPS) (Methods). We observed that pairs of individuals both with ReNU syndrome or both with biallelic

*RNU4-2* variants were significantly closer (or more similar) than pairs of random individuals (linear mixed model $P = 6.4 \times 10^{-54}$ and $P = 1.8 \times 10^{-5}$ for ReNU and biallelic, respectively; Fig. 4b). In contrast, pairs where one individual was ReNU and the other biallelic *RNU4-2* were no closer in CFPS than random pairs ($P = 0.37$). These data indicate there is

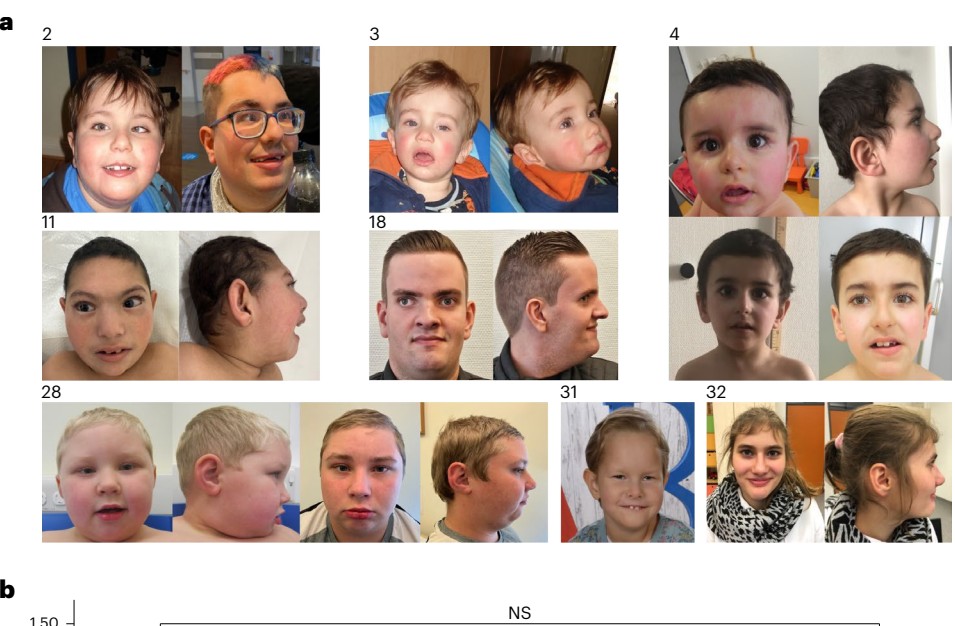

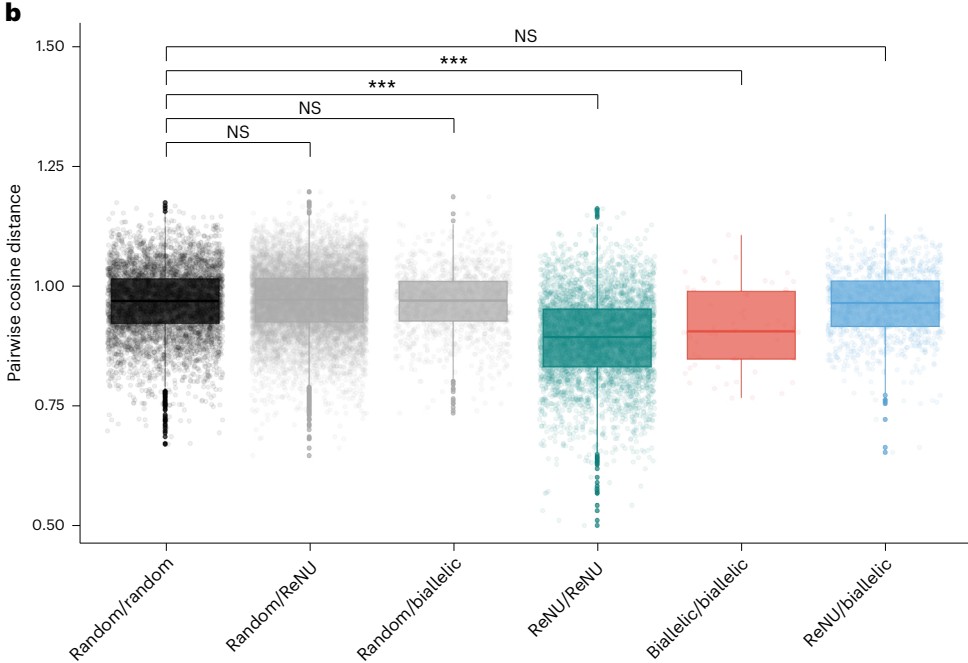

**Fig. 4 | A facial gestalt is associated with biallelic variants in *RNU4-2*. a**, Facial photographs of individuals with biallelic variants identified in *RNU4-2*: individual 2 at 7 and 22 years; individual 3 (sibling of individual 1) at 2 years; individual 4 at 14 months, 4 years (profile and frontal views) and 7 years; individual 11 at 6 years 10 months; individual 18 at 21 years; individual 28 at 5 years 6 months and 14 years 11 months; individual 31 at 7 years; and individual 32 at 17 years. Dysmorphic features were variable across individuals but commonly included a high anterior hairline (seen in individuals 2, 18, 28, 31), synophrys (individuals 2, 4, 11, 18, 32), strabismus (individuals 2, 4, 11, 18, 28, 31), upslanted palpebral fissures (individuals 11, 18, 28, 31), broad nasal bridge and base (individuals 11, 18, 28), thin upper lip (individuals 2, 28, 31) and dental diastema (individuals 2, 11, 28, 31).

**b**, Comparison of facial photographs using GestaltMatcher[15,29]. Pairwise distances were calculated between individuals with ReNU syndrome (*n* = 90), biallelic *RNU4-2* variants (*n* = 11) and random controls with different disorders (*n* = 100). The box plots indicate the median (center) and 25% and 75% quantiles (bounds of the box) of the outcome. The ends of whiskers are determined as the minimum/maximum observation within 1.5 times the IQR from the box. Observations exceeding the whiskers are marked as outliers by points. A linear mixed model was used to compare differences in pairwise distances across groups. NS, not significant; \*\*\**P* < 0.001. Exact *P* values (compared to the random/random group; unadjusted): random/ReNU = 0.570; random/biallelic = 0.827; ReNU/ReNU = 6.44 × 10⁻⁵⁴; biallelic/biallelic = 1.79 × 10⁻⁵; ReNU/biallelic = 0.368.

convergence within, but distinction between, the facial phenotypes of the dominant and recessive disorders.

## Biallelic variants in the *RNU4-2* cluster in important regions revealed by SGE

In the 38 included individuals with recessive *RNU4-2* NDD, we identified 35 unique *RNU4-2* variants, including seven unique homozygous variants in 13 individuals (including three sibling pairs) and 31 unique variants in the compound heterozygous state in 25 individuals (including six sibling pairs) (Fig. 5a,b). Three variants (n.127C>T, n.119A>G and n.7G>C) were observed in both homozygous and compound heterozygous genotypes. While only 20 of 35 (57.1%) variants had significant function scores in the SGE assay (that is, less than −0.302; 5 of 7 (71.4%) homozygous variants and 17 of 31 (54.8%) compound heterozygous variants), the remaining 15 variants were located in the same regions of the U4 structure: nine in the k-turn/5′ stem loop (60.0%), four in Stem II (26.7%) and two in the terminal stem loop (13.3%; Fig. 5a,b). For 13 of 35 (37.1%) of the variants, a variant at the equivalent nucleotide of *RNU4ATAC* was pathogenic

or likely pathogenic in ClinVar or in ref. 14 (including 5 of 15 (33.3%) variants with nonsignificant or absent SGE scores). For 13 further variants (six with nonsignificant or absent SGE scores), there was another variant identified in a different individual either at the same nucleotide (for example, n.7G>A and n.7_8insA) or a pairing nucleotide in the 5′ stem loop of the U4 structure (for example, n.28C>G and n.45G>C; Supplementary Table 7). Furthermore, the location of the UKB variants within the U4 structure differed from those identified in individuals with NDD, with 7 of 14 (50.0%) UKB variants observed in the 3′ stem loop of *RNU4-2* (n.85 to n.117), compared to zero of 35 variants in cases with NDD ($P = 5.01 \times 10^{-4}$, two-sided Fisher's exact test; Fig. 5d).

Using these regional and variant-level annotations, the 35 variants were curated according to the American College of Medical Genetics/Association for Molecular Pathology framework (Methods). Eight variants (22.9%) reached a likely pathogenic classification, one pathogenic (2.86%) and the rest remained a variant of uncertain significance (VUS) (Supplementary Table 7). Ten of 38 individuals (26.3%) had variants on both alleles that were classified as pathogenic or likely pathogenic, four of 38 (10.5%) had a single pathogenic or likely pathogenic allele, and the remaining 24 of 38 individuals (63.2%) had variants on both alleles classified as VUS (Supplementary Table 1). For individuals with detailed clinical information available, there was no difference in the frequency of any phenotypic feature between individuals with two pathogenic or likely pathogenic alleles versus all other individuals (Table 1).

For 13 unrelated individuals with the recessive disorder and two compound heterozygous variants with known inheritance and an SGE score available, we found no difference in mean SGE scores between maternally versus paternally inherited variants (−0.68 and −0.42 respectively; $P = 0.095$, two-sided Mann–Whitney U-test). Similarly, in these individuals, the variant with the strongest SGE score was not significantly more likely to be inherited maternally versus paternally ($P = 0.052$, chi-squared goodness of fit test). We did not observe phenotypic clustering based on variant SGE scores, variant classification or variant position within the U4 secondary structure (Extended Data Fig. 2); however, these analyses are currently underpowered because of our limited sample size.

### Biallelic variants result in loss of *RNU4-2* expression and *RNU4-1* upregulation

We previously showed that individuals with ReNU syndrome have systematic changes in 5′ splice site usage that are detectable in RNA sequencing (RNA-seq) data[7,11]. To characterize whether individuals with biallelic variants have similar splicing changes, we analyzed RNA-seq data for six cases with recessive NDD, three from each of two different studies.

First, we compared jointly processed RNA-seq from lymphocyte cultures for three individuals with biallelic *RNU4-2* variants to 19 individuals with ReNU syndrome and 20 controls with other NDDs. Using the approach from ref. 11, we performed principal components analysis (PCA) on percentage spliced in (PSI) values of exons significantly altered in individuals with ReNU syndrome compared to controls.

Using this approach, the three biallelic *RNU4-2* cases clustered with the controls rather than patients with ReNU syndrome, which is consistent with them not having the same systematic changes to 5′ splice site usage as observed in ReNU syndrome (Fig. 6a).

We next analyzed blood RNA-seq data from 7,826 individuals in the Genomics England National Genomic Research Library (NGRL). This included three individuals with biallelic *RNU4-2* variants, 11 individuals with ReNU syndrome and three individuals who were heterozygous for *RNU4-2* variants with significant SGE scores (that is, function score less than −0.302). Because these transcriptomes were generated using a ribosomal RNA depletion protocol, the expression of noncoding RNAs, including *RNU4-2* and *RNU4-1*, could be quantified across individuals with normalized transcripts per million (TPM). Expression of *RNU4-2* and *RNU4-1* was strongly correlated across samples ($R^2 = 0.75$, $P < 1 \times 10^{-16}$; Fig. 6b and Extended Data Fig. 3). Individuals with biallelic *RNU4-2* variants had a dramatic shift in the *RNU4-2*/*RNU4-1* expression ratio (mean = 0.2) compared to controls (mean = 5.4; two-sided Mann–Whitney U-test $P = 0.003$; Fig. 6c). This was driven both by a strong reduction of *RNU4-2* expression (mean TPM = 314 in biallelic individuals versus 1,645 in controls; $P = 0.012$) and upregulation of *RNU4-1* expression (mean TPM = 1,952 in biallelic individuals versus 346 in controls; $P = 0.003$) in all three samples (Fig. 6c). In three individuals with heterozygous SGE-significant variants, we observed a more moderate reduction in *RNU4-2* levels as would be expected (mean TPM = 420), but no corresponding increase in *RNU4-1*. In contrast, individuals with ReNU syndrome had significantly higher *RNU4-2* levels than controls (mean TPM = 2,699 in ReNU versus 1,645 in controls; $P = 0.023$). Together, these data suggest that dominant and recessive RNU4-2 disorders are driven by different molecular processes and are consistent with a loss-of-function mechanism in the recessive condition.

Finally, we used these data to search for a signature of splicing disruption caused by biallelic variants in *RNU4-2*. In particular, we hypothesized that these variants would lead to an increase in intron retention as observed for variants in the equivalent functional regions of *RNU4ATAC*, which are also thought to act via loss of function. We used IRFinder[16] and DESeq2 (ref. 17) to identify introns that are statistically significant intron retention ratio (IRRatio) outliers in samples from the NGRL. While we saw a clear signal of increased intron retention in U12 (minor) introns in individuals with biallelic *RNU4ATAC* variants, we found no significant intron retention events in either individuals with biallelic variants in *RNU4-2* or individuals with ReNU syndrome (Fig. 6d).

## Discussion

In this study, we characterize the clinical phenotype of a new recessive NDD associated with variants in the *RNU4-2* snRNA gene. We show that this NDD is genetically, phenotypically and mechanistically distinct from ReNU syndrome, which is caused by heterozygous variants within two critical structures in an 18-nucleotide region in the center of *RNU4-2*. Pathogenic variants for the recessive *RNU4-2* condition fall outside the 18-nucleotide ReNU region, instead clustering within

**Fig. 5 | Biallelic variants identified in individuals with NDD cluster in structurally and functionally important regions of *RNU4-2*. a**, Schematic of the U4 snRNA (*RNU4-2*, NR_003137.3) secondary structure in complex with U6. Variants are superimposed on affected nucleotides. **b**, Variants identified in 38 individuals with NDD. Variants are colored according to their function score in the SGE assay[13], which is consistent with **c**. Insertions and deletions not included in the SGE assay are colored in black. The participant number (from Supplementary Tables 1 and 3) associated with each variant, or combination of variants, is given in parentheses. Variants with a significant SGE score (less than −0.302) are prefixed with a single asterisk. **c**, Heatmap of SGE function scores. The minimum SGE score across all SNVs at each position is shown. **d**, Variants observed in UKB compound heterozygous variants are shown from a subset of 200,011 individuals for whom statistical phasing data is available. Heterozygous and homozygous variants are shown for the full cohort of 490,541 genome-sequenced individuals. For heterozygous variants, the height of each ellipse is proportional to the logarithm of the allele count for the most frequent variant at that position (maximum allele count = 1,625); the color represents the SGE score of that variant, which is consistent with **c**. The number of homozygous individuals for each variant is shown. In the full plot, key structural regions are shaded in gray (the 5′ and 3′ stem loop regions and the Stem I region are not shown for clarity). The regions where ReNU syndrome variants occur are shown in teal. Regions important for snRNA–snRNA or snRNA–protein interactions where the biallelic variants cluster occurs are shaded in light gray. Labels for each region are shown above the SGE score heatmap (**c**) and in **a**. Short indels of 1–2 nucleotides are shown at their most 5′ position.

other functionally important elements of *RNU4-2*, including the Stem II region, k-turn region and the Sm protein binding site.

The dominant and recessive *RNU4-2*-associated NDDs have many overlapping phenotypic features, which could indicate a continuum or phenotypic spectrum across the two disorders. However, there is also phenotypic distinction, which is particularly evident in brain imaging. The MRI findings in the recessive *RNU4-2*-associated disorder include cerebellar abnormalities and a prominent white matter phenotype. The most common pattern consists of dilated perivascular spaces, to

varying degrees, in the periventricular and deep white matter regions, often accompanied by corpus callosum and cerebellar atrophy. Cerebellar atrophy involved both the cerebellar hemispheres and vermis. Modern high-resolution MRI has improved detection of dilated perivascular spaces, which in mild forms can be benign[18]. In this cohort, several patients demonstrated very extensive and coalescent dilation of perivascular spaces, producing a tightly packed microcystic appearance. While this severe MRI pattern is characteristic of the recessive *RNU4-2* NDD, other individuals showed milder perivascular enlargement, overlapping

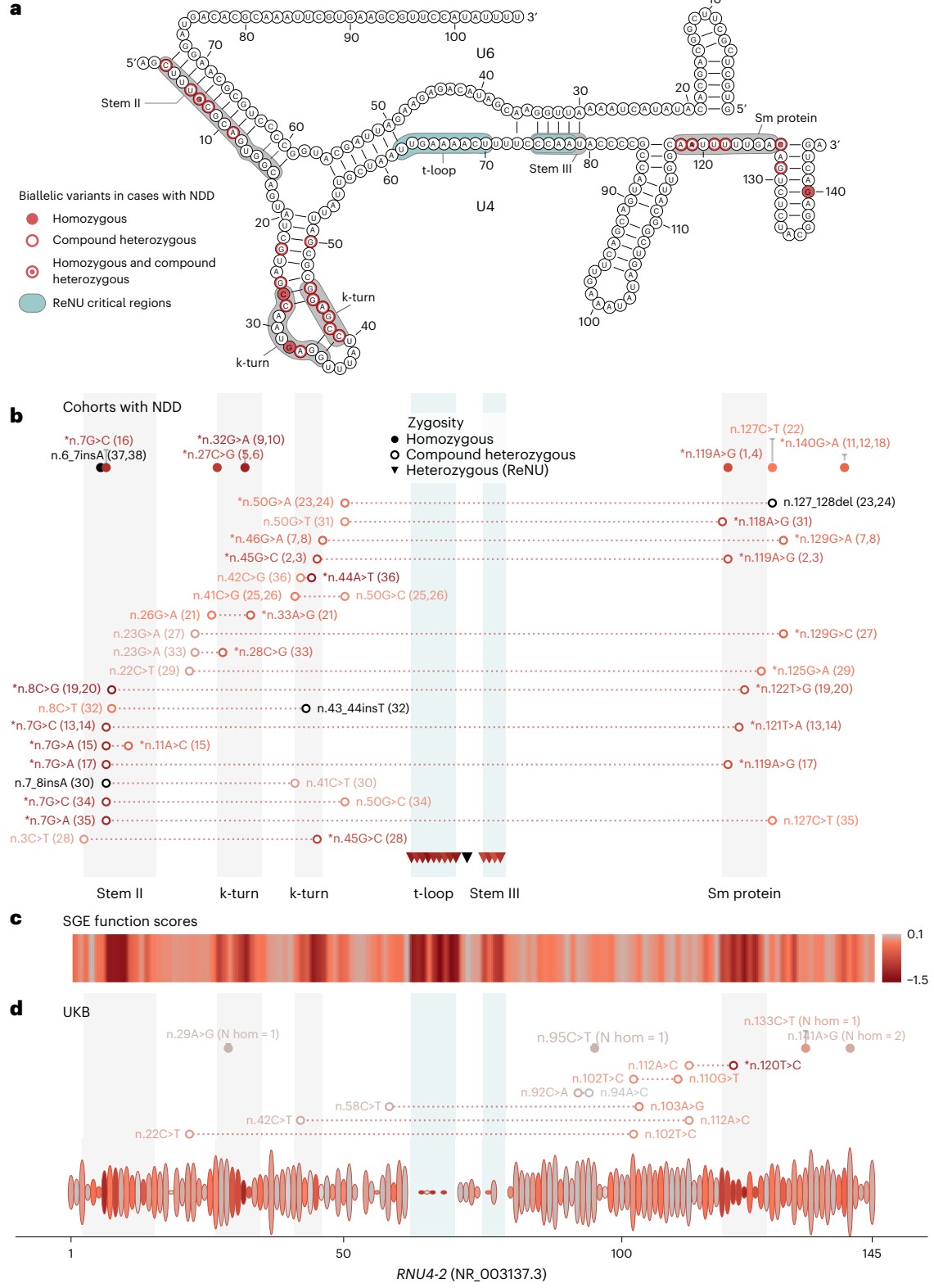

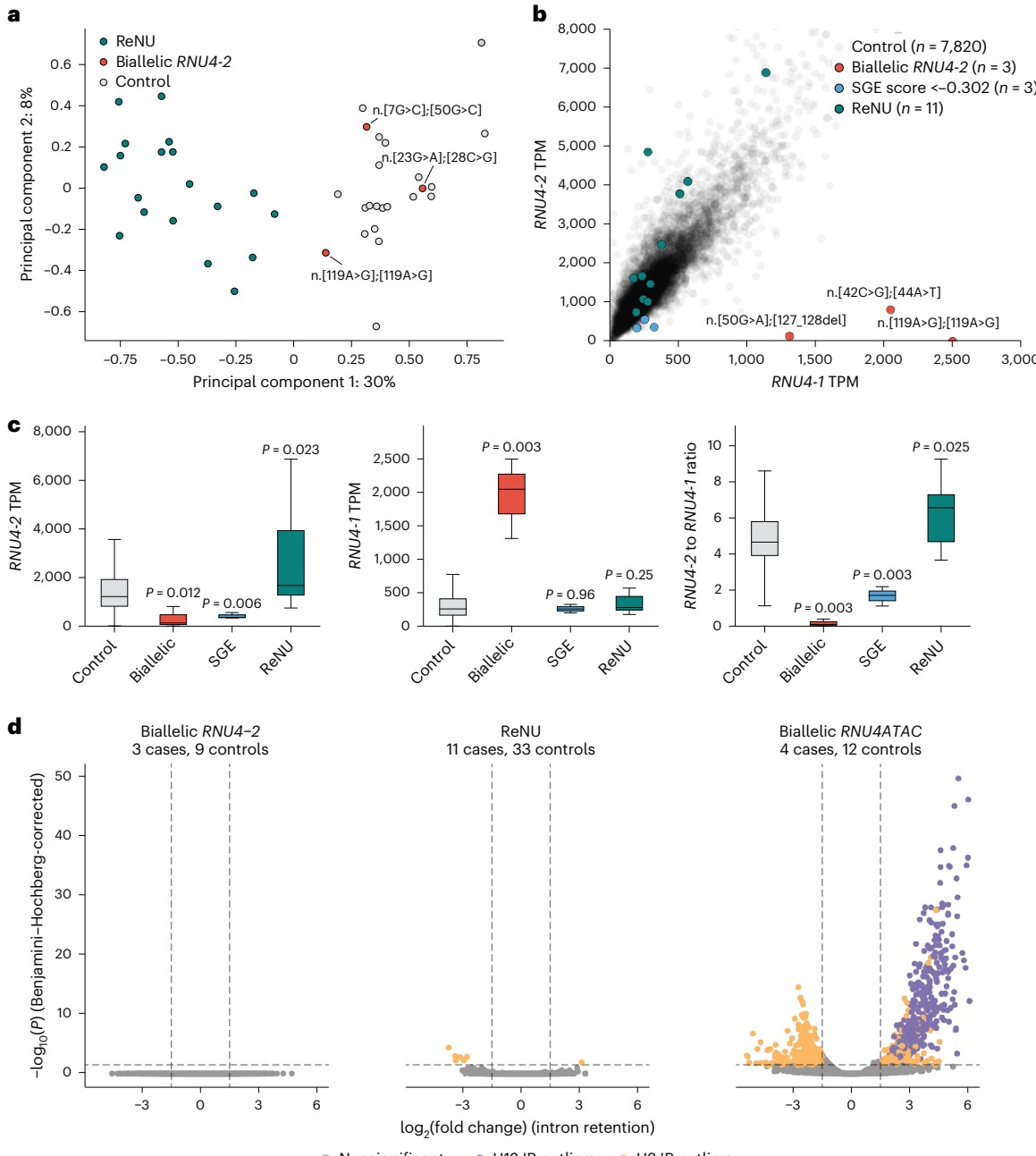

**Fig. 6 | Transcriptomic analysis of individuals with biallelic variants in *RNU4-2* compared to ReNU syndrome. a**, Cases with biallelic variants in *RNU4-2* do not have the ReNU syndrome alternative 5' splice site signature described in ref. 11. Biallelic cases (red) are shown in the principle component space of PSI values of 5' splice sites that are significantly altered in 19 cases with ReNU syndrome (teal) compared to 20 controls (gray). **b**, Expression levels of *RNU4-2* and *RNU4-1* for three individuals with biallelic variants in *RNU4-2* (red), 11 individuals with ReNU syndrome (teal), three control individuals with SGE-significant variants in the heterozygous state (blue) and 7,820 controls with no SGE-significant variants in *RNU4-2* (gray) from blood-derived transcriptomes from Genomics England. **c**, Expression of *RNU4-2* (left),

*RNU4-1* (middle) and their ratio (*RNU4-2* to *RNU4-1*; right) between the groups defined in **b**. *P* values correspond to two-sided Mann–Whitney *U*-tests compared to the control group. Box and whisker plots show the median, quartiles and ± 1.5 times the IQR of the data. **d**, Differentially retained introns identified by IRFinder-Diff in individuals with biallelic *RNU4-2* (left), cases with ReNU syndrome (middle) and *RNU4ATAC* cases (right) versus matched controls. *P* values are from a two-sided Wald test and are Benjamini–Hochberg-corrected to account for multiple comparisons. Introns that are significant outliers in cases and controls are colored according to their type: U12 introns (spliced by the minor spliceosome) are shown in purple; U2 introns (spliced by the major spliceosome) are shown in orange.

with patterns reported in other NDDs[19–21] and metabolic conditions[22] with varying disease mechanisms. The presence of these changes on MRI, particularly the marked periventricular perivascular space dilation, should prompt sequencing and analysis of variants in *RNU4-2*.

Using RNA-seq data, we show that individuals with biallelic variants in *RNU4-2* have dramatically reduced levels of *RNU4-2* RNA, which is consistent with a loss-of-function mechanism. This is accompanied

by an elevation in *RNU4-1* levels, which is consistent with potential compensatory upregulation. The *RNU4-2* to *RNU4-1* ratio discriminates strongly between cases and controls; therefore, it could be a useful diagnostic biomarker for the recessive disorder, if validated in additional cohorts. Our data suggest that *RNU4-1* cannot fully compensate for loss of *RNU4-2*. There could be multiple reasons for this: that *RNU4-1* is functionally distinct from *RNU4-2* in the spliceosome; that *RNU4-1*

cannot be sufficiently upregulated to compensate for *RNU4-2* loss; or that compensation is absent or insufficient in disease-relevant tissues, including the brain. These results contrast with individuals with ReNU syndrome who have marginally elevated rather than reduced *RNU4-2* levels. Similar results were recently described for a new recessive disorder related to *RNU2-2*[23].

Although loss-of-function variants in the equivalent functional regions of *RNU4ATAC* (the minor spliceosomal paralog of *RNU4-2*) to where we identified variants in *RNU4-2* are known to cause intron retention[24,25], we were unable to detect a similar splicing defect in the recessive *RNU4-2* disorder. We suggest that the splicing defect associated with this disorder may be more subtle, as a global defect across U2-type introns is likely to be nonviable. Uncovering the splicing defect in this disorder may require additional samples sequenced at higher depth or sequencing of disease-relevant cell types and tissues.

ReNU syndrome is remarkably prevalent for an NDD, with a frequency similar to many well-known disorders caused by variants in large protein-coding genes[7]. In contrast, recessive *RNU4-2*-associated NDD is much rarer. For example, while we identified 61 individuals with ReNU syndrome in 8,841 individuals (0.69%) with previously undiagnosed NDD in the Genomics England 100,000 Genomes Project, we only identified seven (0.08%) with biallelic variants in the same cohort. However, the relative frequencies of these disorders will differ in populations with high rates of consanguinity, as evidenced by seven individuals (including three sibling pairs) with homozygous variants having consanguineous parents. *RNU4-2*, like other snRNA genes, has a substantially elevated mutation rate[26]. This, combined with negative selection acting on variants across the gene, results in a high density of variants across individuals, but all of these variants remain very rare[7], lowering the chance of homozygous variants arising in populations with low levels of consanguinity. Interestingly, the recently discovered recessive *RNU2-2* disorder[23,27,28] is more common than its dominant counterpart. This may be because the dominant *RNU2-2* NDD is caused by an extremely limited repertoire of de novo variants or because heterozygous variants across *RNU2-2* tend to be seen at higher allele frequencies than those in *RNU4-2*.

We discovered this recessive *RNU4-2*-associated NDD from an SGE experiment, which is detailed in a companion manuscript[13]. Using a stringent significance threshold, we initially identified 20 individuals from 13 families with biallelic variants that were significantly depleted in the SGE assay. In this study, we expanded this cohort to include individuals with variants that do not meet the strict SGE significance threshold. Multiple lines of evidence support our assertion that individuals with SGE-significant and nonsignificant variants have the same recessive *RNU4-2* associated NDD: (1) a consistent phenotype, including observation of the same white matter anomalies on MRI; (2) the localization of variants in each set of individuals to the same structural regions of the U4 snRNA, which differ from the localization of variants in individuals in the UKB; (3) variants with both significant and nonsignificant SGE scores occurring at the equivalent nucleotides as known pathogenic variants in *RNU4ATAC*, supporting that they are disruptive to U4 function.

While the SGE scores are highly predictive for dominant ReNU syndrome variants, they appear less sensitive for recessive disease. This may reflect differences in disease mechanisms: SGE scores measure cell fitness in a haploid cell line and may not capture all relevant aspects of *RNU4-2* function. As such, normal SGE scores should not be used as evidence of benignity in a recessive context. Full calibration of the SGE scores for use in variant classification for the recessive NDD will need to be performed in independent cohorts. In the meantime, we suggest that an evidence strength of 'supporting' should be used for variants that have statistically significant SGE scores.

Without a way to confidently distinguish deleterious from benign variation, it is difficult to ascertain exactly which individuals should be characterized as having the recessive NDD. In this study, we used the distribution of SGE scores of biallelic variants in the UKB to determine a threshold for inclusion, defining a set of individuals with NDD falling outside this range for initial clinical characterization. However, we do not recommend the use of this threshold in clinical settings without further calibration in independent cohorts. The consistent phenotype observed across many members of this cohort suggests that most of these individuals are in fact affected by recessive *RNU4-2* syndrome, but some may be false positives and some of the excluded individuals may also have this recessive NDD. Larger characterized cohorts will be critical to better understand the spectrum of penetrance and variable expressivity for this syndrome and to define criteria to establish variant pathogenicity. Furthermore, we have demonstrated a characteristic MRI phenotype and a molecular RNA signature (the ratio of *RNU4-2* and *RNU4-1* expression) that can be used to identify individuals with this disorder. Future assays using disease-relevant cellular models may also aid the assessment of the functional impact of recessive variants in *RNU4-2*.

In summary, we have characterized the phenotype associated with biallelic variants in *RNU4-2* as a moderate-to-severe syndromic neurodevelopmental disorder with distinct white matter abnormalities. We showed that the recessive NDD is distinct from dominant ReNU syndrome at the genetic, phenotypic and mechanistic levels. These data add to the phenotypic and genotypic spectrum of *RNU4-2*-associated NDDs and highlight the increasing importance of screening snRNA genes to end the diagnostic odyssey for patients with undiagnosed NDD.

## Online content

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

Rocio Rius [1,2,93], Alexander J. M. Blakes [3,4,93], Yuyang Chen [5,6], Joachim De Jonghe [7], François Lecoquierre [5,8], Ruebena Dawes [5,6], Benjamin Cogne [9,10], Hyung Chul Kim [5,6], Javeria R. Alvi[11], Florence Amblard[12,13,14], Morad Ansari[15], Annabelle Arlt[16], Christina Austin-Tse[17,18,19], Sarah Baer[20,21], Meena Balasubramanian [22,23], Elsa V. Balton [24], Giulia Barcia[25,26], Ana Beleza-Meireles[27], Jonathan A. Bernstein [28], Jasmin Beygo[29], Pierre Blanc[30], Nuria C. Bramswig [31], Frederik Braun [29], Daniel Buchzik[32], Daniel G. Calame[33,34], Jamie Campbell[15], Charles Coutton[12,13,14], Chloe A. Cunningham [35,36], Nitsuh Dargie[24], Christel Depienne [29], Katrina M. Dipple[37,38], Anne Dieux[39,40], Abhijit Dixit [41], Lauren Dreyer[42], Haowei Du[43], Salima El Chehadeh [44,45], Michael Field [46], Lisa J. Ewans[47,48,49], Vanessa Geiger[50], Richard A. Gibbs[43,51], Ian Glass [37,38], Olivier Grunewald[52,53], Paul Gueguen[30,54,55], Tobias B. Haack [56], Hamza Hadj Abdallah[25,26], Radu Harbuz[12,13,14], Ingo Helbig [57,58,59,60], Judit Horvath[31], Alexander Hustinx [16], Bertrand Isidor[61], Marie-Line Jacquemont[54,55,62], Fraser Jamie[63], Médéric Jeanne[54,55,62,64], Riley Kessler[59], Hannah Klinkhammer [16,65], G. Christoph Korenke[66], Urania Kotzaeridou[67], Peter Krawitz [16], Steven Laurie [68,69], Richard J. Leventer [36,70,71], Rebecca J. Levy [72], James R. Lupski [34,43,51,73], Pierre Marijon [30], Kaitlin E. McGinnis[42], Rodrigo Mendez[74], Olfa Messaoud [17,18,75,76], Caroline Nava [30,77], Mevyn Nizard[25,26,78], Anne O'Donnell-Luria[17,18,75,79], Melanie C. O'Leary [17], Simone Olivieri[56], Amitav Parida[80], Davut Pehlivan[33,34], Anna Jenne Prentice [57,59], Jennifer E. Posey [43], Chloe M. Reuter[28,74], Véronique Satre[12,13,14], Caroline Schluth-Bolard[13,81], Thomas Smol[39], Tipu Sultan[11], John Taylor[82], Christel Thauvin-Robinet [83], Julien Thevenon[12,13,14], Eloise Uebergang [70], Sandra Ueberberg[29], Catherine Vincent-Delorme[40,84], Evangeline Wassmer [85,86], Emma Westwood[87], Matthew T. Wheeler [74], Elif Yilmaz Gulec[88,89], Adeline Vanderver[59,60], Arastoo Vossough[90], Stephan J. Sanders [91,92], Siddharth Banka [3,4], Gregory M. Findlay [7], Daniel G. MacArthur[1,2], Cas Simons [1,2,94] ✉ & Nicola Whiffin [5,6,17,94] ✉

[1]Centre for Population Genomics, Garvan Institute of Medical Research, UNSW Sydney, Sydney, New South Wales, Australia. [2]Centre for Population Genomics, Murdoch Children's Research Institute, Melbourne, Victoria, Australia. [3]Manchester Centre for Genomic Medicine, Division of Evolution and Genomic Sciences, School of Biological Sciences, Faculty of Biology, Medicine and Health, University of Manchester, Manchester, UK. [4]Manchester Centre for Genomic Medicine, St Mary's Hospital, Manchester University NHS Foundation Trust, Health Innovation Manchester, Manchester, UK.

[5]Big Data Institute, University of Oxford, Oxford, UK. [6]Centre for Human Genetics, University of Oxford, Oxford, UK. [7]The Genome Function Laboratory, The Francis Crick Institute, London, UK. [8]Univ Rouen Normandie, Inserm U1245 and CHU Rouen, Department of Genetics and Reference Center for Developmental Abnormalities, Rouen, France. [9]Nantes Université, CHU de Nantes, CNRS, INSERM, l'institut du thorax, Nantes, France. [10]Nantes Université, CHU de Nantes, CNRS, INSERM, Génétique médicale, Nantes, France. [11]Department of Pediatric Neurology, University of Child Health Sciences, The Children's Hospital, Lahore, Pakistan. [12]Service de Génétique, Génomique et Procréation, CHU Grenoble Alpes, Grenoble, France. [13]GCS AURAGEN, Lyon, France. [14]Université Grenoble Alpes, INSERM U 1209, CNRS UMR 5309, Institut for Advanced Biosciences, Grenoble, France. [15]South East Scotland Clinical Genetics Service, NHS lothian, Edinburgh, UK. [16]Institute for Genomic Statistics and Bioinformatics, Medical Faculty, University of Bonn, Bonn, Germany. [17]Center for Mendelian Genomics, Program in Medical and Population Genetics, Broad Institute of MIT and Harvard, Cambridge, MA, USA. [18]Center for Genomic Medicine, Massachusetts General Hospital, Boston, MA, USA. [19]Department of Pathology, Harvard Medical School, Boston, MA, USA. [20]Department of Neuropediatrics, ERN EpiCare, French Centre de référence des Épilepsies Rares (CréER), Hôpitaux Universitaires de Strasbourg, Strasbourg, France. [21]Institute for Genetics and Molecular and Cellular Biology (IGBMC), University of Strasbourg, CNRS UMR7104, Illkirch, France. [22]Division of Clinical Medicine, School of Medicine and Population Health, University of Sheffield, Sheffield, UK. [23]Sheffield Clinical Genomics Service, Sheffield Children's NHS Foundation Trust, Sheffield, UK. [24]Department of Medicine, University of Washington School of Medicine, Seattle, WA, USA. [25]Genomic Medecine of Rare Disease, Necker Hospital, Paris, France. [26]Imagine Institute, Paris, France. [27]Clinical Genetics Department, Guy's and St Thomas' NHS Foundation Trust, London, UK. [28]Department of Pediatrics, Stanford University School of Medicine, Stanford, CA, USA. [29]Institute of Human Genetics, University Hospital Essen, University Duisburg-Essen, Essen, Germany. [30]Laboratoire SeqOIA, Paris, France. [31]Department of Medical Genetics, Centre of Medical Genetics, University and University Hospital Münster, Münster, Germany. [32]Department of Neuropediatrics, Diak Klinikum Landkreis Schwäbisch Hall, Schwäbisch Hall, Germany. [33]Section of Pediatric Neurology, Department of Pediatrics, Baylor College of Medicine, Houston, TX, USA. [34]Texas Children's Hospital, Houston, TX, USA. [35]Victorian Clinical Genetics Services, Murdoch Children's Research Institute, Melbourne, Victoria, Australia. [36]Department of Paediatrics, University of Melbourne, Melbourne, Victoria, Australia. [37]Department of Pediatrics, University of Washington, Seattle, WA, USA. [38]Brotman Baty Institute for Precision Medicine, Seattle, WA, USA. [39]CHU Lille, ULR7364 – RADEME – Maladies Rares du Développement Embryonnaire, Lille, France. [40]Clinique de Génétique, Hôpital Jeanne de Flandre, CHU de Lille, Lille, France. [41]Clinical Genetics, Nottingham University Hospitals, Nottingham, UK. [42]Genetic Health WA, Perth, Western Australia, Australia. [43]Department of Molecular and Human Genetics, Baylor College of Medicine, Houston, TX, USA. [44]Service de Génétique Médicale, Institut de Génétique Médicale D'Alsace, Hôpitaux Universitaires de Strasbourg, Strasbourg, France. [45]Laboratoire de Génétique Médicale, Institut de Génétique Médicale d'Alsace, INSERM UMRS_1112, CRBS, Université de Strasbourg, Strasbourg, France. [46]Genetics of Learning Disability Service, Hunter Genetics, Waratah, Western Australia, Australia. [47]Centre for Clinical Genetics, Sydney Children's Hospitals Network, Sydney, New South Wales, Australia. [48]Genomics and Inherited Diseases Program, Garvan Institute of Medical Research, Darlinghurst, New South Wales, Australia. [49]Discipline of Paediatrics and Child Health, School of Clinical Medicine, Faculty of Medicine and Health, University of New South Wales, Sydney, New South Wales, Australia. [50]Genetikum, MVZ genetikum GmbH, Neu-Ulm, Germany. [51]Human Genome Sequencing Center, Baylor College of Medicine, Houston, TX, USA. [52]U1172-LilNCog-Lille Neuroscience & Cognition, CHU de Lille, Lille, France. [53]Laboratoire de Genopathies, CHU Lille, Lille, France. [54]Service de Génétique, CHRU de Tours, Tours, France. [55]Université de Tours, INSERM, Imaging Brain & Neuropsychiatry iBraiN, Tours, France. [56]Institute of Medical Genetics and Applied Genomics, Eberhard Karls University, Tübingen, Germany. [57]The Epilepsy NeuroGenetics Initiative (ENGIN), Children's Hospital of Philadelphia, Philadelphia, PA, USA. [58]Epilepsy and Neurodevelopmental Disorders Center (ENDD), Children's Hospital of Philadelphia, Philadelphia, PA, USA. [59]Division of Neurology, Department of Pediatrics, Children's Hospital of Philadelphia, Philadelphia, PA, USA. [60]Department of Neurology, Perelman School of Medicine, University of Pennsylvania, Philadelphia, PA, USA. [61]Service de Génétique Médicale, Institut de Génétique Médicale D'Alsace, Centre Hospitalier Universitaire de Nantes, Nantes, France. [62]Centre de Référence Maladies Rares "Anomalies du Développement et Syndromes Malformatifs", FHU Genomeds, CHRU de Tours, Tours, France. [63]Rare Disease Institute, Division of Genetics and Metabolism and Center for Genetic Medicine Research, Children's National Hospital, Washington, DC, USA. [64]PRISME division for congenital and Developmental Disorders, Department of Genetics, Hôpital de l'Estran, Avranches, France. [65]Institute for Medical Biometry and Statistics, Marburg University, Marburg, Germany. [66]Department of Neuropediatrics, University Children's Hospital, Klinikum Oldenburg, Oldenburg, Germany. [67]Department of Pediatrics I, Division of Pediatric Neurology and Metabolic Medicine, Medical Faculty of Heidelberg, Heidelberg, Germany. [68]Centro Nacional de Análisis Genómico (CNAG), Baldiri Reixac 4, Barcelona, Spain. [69]Universitat de Barcelona (UB), Barcelona, Spain. [70]Murdoch Children's Research Institute, Melbourne, Victoria, Australia. [71]Royal Children's Hospital, Melbourne, Victoria, Australia. [72]Division of Child Neurology, Department of Neurology and Neurological Sciences, Stanford University, Stanford, CA, USA. [73]Department of Pediatrics, Baylor College of Medicine, Houston, TX, USA. [74]Cardiovascular Medicine, Stanford University, Stanford, CA, USA. [75]Division of Genetics and Genomics, Boston Children's Hospital, Boston, MA, USA. [76]Harvard Medical School, Boston, MA, USA. [77]Sorbonne Université, Institut du Cerveau - Paris Brain Institute - ICM, Inserm, CNRS, APHP, Département de Génétique, Hôpital de la Pitié Salpêtrière, Paris, France. [78]Paris Cité University, Paris, France. [79]Department of Pediatrics, Harvard Medical School, Boston, MA, USA. [80]Department of Paediatric Neurology, Birmingham Women's and Children's Hospital Foundation Trust, Birmingham, UK. [81]Laboratoire de Diagnostic Génétique, Institut de Génétique Médicale d'Alsace, INSERM UMRS_1112, Université de Strasbourg, Hôpitaux Universitaires de Strasbourg, Strasbourg, France. [82]Department of Radiology, NHS lothian, Edinburgh, UK. [83]Université Bourgogne Europe - CHU Dijon Bourgogne - Inserm U1231 CTM GAD, Centre de Référence des maladies neurogénétiques, Laboratoire de Génomique Médicale, Dijon, France. [84]Consultation de génétique, CH Arras, Arras, France. [85]Birmingham Children's Hospital, Birmingham, UK. [86]Institute of Health and Neurodevelopment, Aston University, Birmingham, UK. [87]NHS Education for Scotland, NHS Scotland, Edinburgh, UK. [88]Department of Medical Genetics, Istanbul Medeniyet University Medical School, Istanbul, Turkey. [89]Medical Genetics Clinic, Istanbul Goztepe Prof Dr Suleyman Yalcin City Hospital, Istanbul, Turkey. [90]Department of Radiology, Children's Hospital of Philadelphia, Philadelphia, PA, USA. [91]Institute of Developmental and Regenerative Medicine, Department of Paediatrics, University of Oxford, Oxford, UK. [92]Department of Psychiatry and Behavioral Sciences, UCSF Weill Institute for Neurosciences, University of California San Francisco, San Francisco, CA, USA. [93]These authors contributed equally: Rocio Rius, Alexander J. M. Blakes. [94]These authors jointly supervised this work: Cas Simons, Nicola Whiffin. ✉e-mail: cas.simons@populationgenomics.org.au; nwhiffin@well.ox.ac.uk

## Methods

### NDD cohort and clinical data collection

We searched rare disease cohorts for individuals with biallelic variants in *RNU4-2* and undiagnosed neurodevelopmental phenotypes. These cohorts included the Genomics England 100,000 Genomes Project and NHS Genomic Medicine Service datasets accessed through the UK NGRL[30], the Center for Population Genomics CaRDinal cohort, the SeqOIA and Auragen clinical cohorts in France (Plan France Medicine Genomics 2025 (PFMG 2025); https://pfmg2025.fr/en/), the Undiagnosed Disease Network, the Broad Institute Center for Mendelian Genomics and Genomics Research to Elucidate the Genetics of Rare Diseases (GREGoR)[31] Consortium cohorts. Individuals were excluded if the variants did not segregate with NDD in the family. Variants were excluded if they were observed as homozygous in either UKB or All of Us. Additional individuals were identified through personal communications. For individuals recruited as trios, variants were phased using parental sequencing data. For individuals with two variants in *RNU4-2* but without sequencing data from one or both parents, variant phasing was manually determined by inspection of reads in the Integrative Genomics Viewer[32].

Informed consent was obtained for all patients included in this study from their parent(s) or legal guardian, with the study approved by the local regulatory authority. A specific consent form was obtained from the families who consented to the publication of photographs. The 100,000 Genomes Project Protocol has ethical approval from the Health Research Authority Committee East of England Cambridge South (Research Ethics Committee ref. no. 14/EE/1112). This study was registered with Genomics England under Research Registry Projects 354. Health-related research in the UKB was approved by the Research Ethics Committee under ref. no. 16/NW/0274, with this research conducted under application number 81050.

Clinical collaboration requests were submitted to Genomics England to contact recruiting clinicians and collect additional phenotypic information. Clinical data were collected and summarized for features seen across the cohort. Written informed consent was obtained to publish all photographs and MRI images.

### Comparison of recessive phenotypes with dominant ReNU syndrome

For a subset of 27 individuals in this cohort for whom detailed phenotypic information was available (Supplementary Table 3), we counted the number of individuals in whom each phenotype was present or definitely absent.

We obtained the same information for ReNU syndrome by combining counts from Table 1 and Supplementary Table 2 of ref. 7 and Supplementary Table 7 of ref. 11 given that the authors of ref. 11 took care to ensure that the two cohorts were nonoverlapping. We included only phenotypes that were present in at least 25% of individuals with biallelic variants from this study, or at least 25% of individuals in the combined ReNU cohorts. Details of the precise phenotypic terms aggregated across the cohorts are given in Supplementary Table 5. We then calculated the odds ratio (with Haldane–Anscombe correction) for the presence of each phenotype in this cohort versus the combined ReNU cohorts. Statistical significance was determined using two-sided Fisher's exact tests, followed by FDR correction (Benjamini–Hochberg method).

### Phenotypic clustering analysis

We reproduced the PCA described in ref. 11. For individuals with detailed clinical phenotyping data, HPO terms were encoded as present (1) or absent (0). Missing data were annotated as absent. ID was further stratified into mild (1), moderate (2) or severe (3) categories. PCA was performed on these encoded data. Only one individual from each sibling pair was included in this analysis.

### Annotation of biallelic variants in cases with NDD and population controls

We identified variants in *RNU4-2* from short-read genome sequencing data in 490,541 individuals from the UKB[33] (DRAGEN pipeline) and in 414,840 individuals from All of Us version 8. We additionally identified individuals with compound heterozygous variants in a subset of 200,011 UKB participants with statistical phasing information (https://biobank.ndph.ox.ac.uk/crystal/ukb/docs/PhasingUKB200k_report_SHAPEIT.pdf).

SGE scores were taken from ref. 13. For each nucleotide position in *RNU4-2*, we calculated the minimum SGE score of any SNV at that position. To compare individuals with NDD to those in the UKB, for each individual with biallelic variants in *RNU4-2*, we calculated the mean SGE score across their two alleles.

For insertions and deletions without available SGE data, the function score was inferred from the mean SGE score across all SNVs within the deleted nucleotides or taking the mean SGE score across all SNVs within the nucleotides directly flanking the insertion. To validate this approach, we compared the SGE function score of 70 tested single-base insertions and eight tested single-base deletions to the score that would be inferred using this approach. The experimental and inferred scores were strongly correlated (Spearman rank correlation coefficient = 0.83, $P < 0.001$ for insertions, and 0.83, $P = 0.015$ for deletions; Extended Data Fig. 4a). For most of the insertions and deletions that differed in their inferred and experimental scores by more than 0.2 (22 of 33; 66.7%), the inferred score underestimated the deleteriousness of the variant (Extended Data Fig. 4b), which is consistent with indels having a more severe effect than SNVs. Only four of the 78 variants (5.1%) had inferred scores that would cross the threshold of significance (less than −0.302) but had a measured score below that threshold.

Variants were annotated with the region of *RNU4-2* to which they map using the following nucleotides: Stem II (n.3 to n.16); 5′ stem loop (n.20 to n.52); Stem I (n.56 to n.62); t-loop (n.63 to n.70); Stem III (n.75 to n.79); 3′ stem loop (n.85 to n.117); Sm protein (n.118 to n.126); and terminal stem loop (n.127 to n.144). Within the 5′ stem loop, the k-turn was annotated as n.27 to n.35 and n.41 to n.46.

Two regions of *RNU4-2* and *RNU4ATAC* with identical structures were defined as follows: *RNU4-2* n.26 to n.52 with *RNU4ATAC* n.31 to n.57, and *RNU4-2* n.115 to n.126 with *RNU4ATAC* n.113 to n.124. Variants at the same nucleotide in the structure and where the reference bases in *RNU4-2* and *RNU4ATAC* are identical, were marked as 'equivalent'. Within the 5′ stem loop of *RNU4-2*, pairing nucleotides were determined as in Fig. 4 of ref. 13.

### Variant classification

All identified variants were classified using the established framework from the American College of Medical Genetics and Genomics and Association for Molecular Pathology[34] and additional specifications for noncoding variants[35]. 'PM2 supporting' was applied for variants rare in the UKB (allele frequency < 0.1% and no homozygotes). 'PS3 supporting' was applied to variants with significant SGE function scores, with the evidence level capped at supporting because of the absence of known pathogenic and benign variation to properly benchmark this assay for biallelic *RNU4-2* variants. 'PM1' was applied to variants in Stem II, the k-turn or the Sm protein site (see section above). 'PM3' was applied according to updated guidance from ClinGen (https://clinicalgenome.org/site/assets/files/3717/svi_proposal_for_pm3_criterion_-_version_1.pdf) but with no evidence given to variants in *trans* with variants of uncertain significance given the high variant density across *RNU4-2*. 'PM5' was applied to variants with an exact equivalent variant in *RNU4ATAC* classified as pathogenic or likely pathogenic in ClinVar. 'PP4' was added for variants found in patients where a loss of *RNU4-2* expression and increase in *RNU4-1* expression was observed through RNA-seq (Fig. 6).

## GestaltMatcher analysis of facial photographs

GestaltMatcher is an artificial-intelligence-driven facial pheno-typing tool trained on images of patients with one of 274 different Mendelian disorders from the GestaltMatcher Database (GMDB) (https://db.gestaltmatcher.org/)[15,29]. Each image is encoded by twelve 512-dimensional representation vectors spanning a CFPS. Facial simi-larities between two images can be quantified by the cosine distance between their respective CFPS representations.

To assess facial similarity among ReNU and biallelic individuals, phenotypic distances in the CFPS of GestaltMatcher[15,36] were analyzed. To ensure an unbiased assessment, ReNU cases were excluded from the training of GestaltMatcher. Then, three groups were considered: (1) 100 individuals from the GMDB that have not been included in the training of GestaltMatcher and that have been diagnosed with 100 dif-ferent disorders ('random'); (2) 90 ReNU individuals from the GMDB ('ReNU'); and (3) 11 biallelic individuals ('biallelic'). For each individual, one facial image (portrait) was used. For each pair of individuals, the pairwise cosine distance of their CFPS representations was derived. Comparisons of related individuals (siblings) and outliers (mean pair-wise cosine distance less than 0.5) were excluded.

A linear mixed model was used to compare differences in pairwise distances across groups. Pairwise distance was used as the depend-ent variable and group as the independent variable, where 'random/random' was defined as the reference group. As each individual was included in several comparisons, random intercepts were used for both individual identifications in the comparisons.

## Testing for the ReNU 5′ splice site signature in individuals with biallelic *RNU4-2* variants

RNA-seq from cultured lymphocytes was performed according to the protocol described in ref. 11 for *RNU4-2*. rMATS-turbo (v.4.3.0)[37] was run on 19 ReNU samples and 20 controls; 101 significant A5SS events (FDR < 0.1, deltaPSI > 0.05) were retained. Then, rMATS-turbo was rerun on the 19 ReNU samples, the 20 controls and three biallelic RNU4-2 samples, with-out statistical or deltaPSI filtering. The A5SS output was filtered on the 101 retained events and the PSI values were extracted to perform the PCA.

## *RNU4-2* and *RNU4-1* expression analysis

We analyzed the expression profile of *RNU4-2* and *RNU4-1* using tran-scriptomic data from The Genomics England 100kGP Transcriptomics Pilot and Extension (https://re-docs.genomicsengland.co.uk/rna_seq/). This dataset includes whole-blood ribodepletion transcriptomes of 7,840 samples from 7,829 participants with rare diseases. Three sam-ples exhibiting outlier read counts or lacking quality control informa-tion, as noted in the Genomics England documentation (https://re-docs.genomicsengland.co.uk/rna_seq_pilot/), were excluded, yielding a final dataset of 7,837 samples from 7,826 participants. Read alignment and transcript quantification were performed using the DRAGEN RNA Pipeline v.3.8.4 and v.4.2.7, with annotations from gencode v.32. We extracted quantifications of *RNU4-2* (ENSG00000202538.1) and *RNU4-1* (ENSG00000200795.1). The correlation between *RNU4-2* and *RNU4-1* expression levels was assessed using ordinary least squares linear regression on $\log_{10}$-transformed values ($\log_{10}$(*RNU4-2*) - $\log_{10}$(*RNU4-1*)).

## Analysis of intron retention in RNA-seq data

We used RNA-seq data from three individuals with biallelic *RNU4-2* variants, 11 individuals with ReNU syndrome caused by the recurrent insertion (n.64_65insT) and four individuals with biallelic *RNU4ATAC* variants. Individuals with *RNU4ATAC* diagnoses were identified in the NGRL exit questionnaire table ('case solved family' with *RNU4ATAC* listed as the gene name) and diagnostic discovery table (with gene name 'RNU4ATAC'). Identified individuals were manually checked for biallelic *RNU4ATAC* variants.

For each of the above samples, we selected three control RNA-seq samples matched on age at consent, sex, ancestry and mil mapped reads.

Controls were selected from a set of RNA-seq samples in Genomics England with mil mapped reads > 60 and excluding individuals with normalized disease group = 'neurology and neurodevelopmental disorders'.

RNA-seq BAM files were input into IRFinder (v.2.0.1)[16] with GENCODE v.46 as a reference. IRFinder 'Diff' mode was then run with DESeq2[17] enabled, separately for each case and matched con-trol set, to identify outliers for intron retention in each group. The DESeq2 wrapper fits a generalized linear model using intronic and spliced exonic reads as quantified by the IRFinder program, and tests the fold change and Benjamini–Hochberg-corrected sig-nificance of intron retention between two groups of samples. U12 introns were annotated using a list of 810 U12 introns retrieved from Supplementary Table 8 of ref. 38.

## Reporting summary

Further information on research design is available in the Nature Portfolio Reporting Summary linked to this article.

## Data availability

Data from the NGRL used in this research are available within the secure Genomics England Research Environment. Access to NGRL data is restricted to adhere to consent requirements and protect participant privacy. Access to NGRL data is provided to approved researchers who are members of the Genomics England Research Network, subject to institutional access agreements and research project approval under participant-led governance. For more infor-mation on data access, visit: www.genomicsengland.co.uk/research. Genomic and phenotypic data from the GREGoR consortium (includ-ing the Rare Genomes Project cohort) and the Undiagnosed Disease Network are available through dbGaP accession nos. phs003047. v1.p1 and phs001232.v5.p2, respectively, with at least annual data releases. Access is managed by a data access committee designated by dbGaP and is based on the intended use of the requester and allowed use of the data submitter as defined by consent codes. Data access to individual genome data from PFMG 2025 with other research-ers is subject to current data protection and regulations in France and is only possible through the Collecteur Analyseur de Données (CAD). More information on data access and the CAD structure can be obtained on the PFMG 2025 website (https://pfmg2025.fr/le-plan/collecteur-analyseur-de-donnees-cad/). UKB data are avail-able at the Research Analysis Platform for use by eligible researchers under approved access. This research was conducted under applica-tion no. 81050. The RNA-seq data in Fig. 6a are available at the European Genome-phenome Archive (accession no. EGAS50000000889). Data from Fig. 6b–d are available within the NGRL (see access details above). Variant curations have been deposited in ClinVar under submission no. SUB16002064. All other data are available within the paper and Supplementary Tables 1–7.

## Code availability

This study predominantly used available software packages: bedtools (v.2.31.0), bcftools (v.1.16) and samtools (v.1.9). R (v.4.1.1) was used via RStudio with plots generated using ggplot (v.3.5.2) and related pack-ages. The code generated for the analyses within this manuscript is stored securely in the NGRL and can be shared within this environment upon request.

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

## Acknowledgements

We thank all the families with rare diseases who kindly contributed their genomic and clinical data to make this work possible. We thank P. O'Donovan, M. Sato and E. Miller from the Genomics England Airlock team. N.W. is supported by a Wellcome Career Development Award (grant no. 305292/Z/23/Z), a Lister Institute research prize and grant funding from Novo Nordisk. Y.C. is supported by a studentship from Novo Nordisk. The Francis Crick Institute receives its core funding (G.M.F.) from Cancer Research UK (CC2190), the UK Medical Research Council (CC2190) and the Wellcome Trust (CC2190). A.B. is supported by a Wellcome PhD Training Fellowship for Clinicians and the 4Ward North PhD Programme for Health Professionals (223521/Z/21/Z). The analysis was supported by the Centre for Population Genomics (Garvan Institute of Medical Research and Murdoch Children's Research Institute) (D.G.M. and C.S.) and was funded in part by a National Health and Medical Research Council investigator grant (2009982) and the Medical Research Future Fund Genomics Health Futures Mission (2032931). Massimo's Mission acknowledges funding support from the Australian Government Department of Health and Aged Care (EPCD000034) (D.G.M. and C.S.). O.M. is supported by the Hazem Ben-Gacem Tunisia Medical Fellowship Fund. R.J.L. is supported by the CNCDP-K12 and an NINDS K08 for research unrelated to this publication. T.B.H. received funding from the European Commission (Recon4IMD–AP-101080997) and the German Research Foundation (EJP-RD Artemis: 542553983). D.G.C. was supported by the National Institute of Neurological Disorders and Stroke of the National Institutes of Health under award no. K12NS098482. S.B. acknowledges the support of the MRC Epigenomics of Rare Diseases (EpiGenRare) Node (MR/Y008170/1). This study was supported by the National Institute for Health and Care Research (NIHR) Manchester Biomedical Research Centre (NIHR203308 to S.B.) and funded in part by the US National Institutes of Health Genomics Research Elucidates Genetics of Rare, GREGoR, Program (no. U01HG011758 to E.P., J.R.L. and R.A.G.; no. U01HG011755 to A.O'D.-L., M.C.O'L.). The sequencing and analysis of individual no. 24 were provided by the Broad Institute Center for Mendelian Genomics (A.O'D.-L., M.C.O'L.) and were funded by the National Human Genome Research Institute (nos. U01HG011755 (GREGoR consortium), R01HG009141), and in part by the Chan Zuckerberg Initiative Donor-Advised Fund at the Silicon Valley Community Foundation (funder DOI 10.13039/100014989) grant nos. 2020-224274, 2022-309464, 2022-316726 and 2022-316726 (https://doi.org/10.37921/236582yuakxy). The Solve-RD project has received funding from the European Union's Horizon 2020 research and innovation programme under grant no. 779257 (S.L.). The research reported in this publication was supported by the National Institute Of Neurological Disorders and Stroke of the National Institutes of Health (no. U01NS134358 to R.M., M.T.W., J.A.B. and C.M.R.; no. U01NS134355 to K.M.D. and E.V.B.). N.C.B. and J.H. are members of the European Reference Network for Developmental Anomalies and Intellectual Disability (ERN-ITHACA). The content is solely the responsibility of the authors and does not necessarily represent the official views of the funding agencies. We thank the participants of the NGRL, whose contributions made this research possible. Secure access to the NGRL under project no. 354 was provided by Genomics England, which delivers the NGRL in partnership with NHS England and is wholly owned by the UK Department of Health and Social Care. The NGRL contains participants' health data collected by the NHS as part of their care, along with samples and data from their participation in research, for which fully informed consent has been obtained. This includes the genomic and clinical data provided through the NHS Genomic Medicine Service, as well as data obtained through research studies, including the 100,000 Genomes Project and the Generation Study, both of which are delivered in partnership with the NHS, and from other research cohorts involving external collaborators. We thank the All of Us and UKB participants for their contributions. We also thank the National Institutes of Health's All of Us Research Program for making available the participant and variant data examined in this study. For the purpose of open access, the authors have applied a CC BY public copyright license to any author accepted manuscript version arising from this submission.

## Author contributions

R.R., A.J.M.B., Y.C., J.D.J., F.L., R.D., B.C. and H.C.K. performed the data analysis. R.R., A.J.M.B., R.D. and F.L. prepared the figures. A. Vanderver and A. Vossough analyzed the MRI images. S.J.S., S. Banka, G.M.F., D.G.M., C.S. and N.W. supervised the work. J.R.A., F.A., M.A., A.A., C.A.-T., S.Baer, M.B., E.V.B., G.B., A.B.-M., J.A.B., J.B., P.B., N.C.B., F.B., D.B., D.G.C., J.C., C.C., C.A.C., N.D., C.D., K.M.D., A.Dieux, A.Dixit, L.D., H.D., S.E.C., M.F., L.J.E., V.G., R.A.G., I.G., O.G., P.G., T.B.H., H.H.A., R.H., I.H., J.H., A.H., B.I., M.-L.J., F.J., M.J., R.K., H.K., G.C.K., U.K., P.K., S.L., R.J.Leventer, R.J.Levy, J.R.L., P.M., K.E.M., R.M., O.M., C.N., M.N., A.O'D.-L., M.C.O'L., S.O., A.P., D.P., A.J.P., J.E.P., C.M.R., V.S., C.S.-B., T. Smol, T. Sultan, J. Taylor, C.T.-R., J. Thevenon, E.U., S.U., C.V.-D., E. Wassmer, E. Westwood, M.T.W. and E.Y.G. provided access to the patient phenotype data, the MRI images or photographs. R.R., A.J.M.B., D.G.M., C.S. and N.W. co-wrote the manuscript. All other authors read the manuscript and provided comments.

## Competing interests

N.W. receives research funding from Novo Nordisk and BioMarin Pharmaceutical. D.G.M. is a paid consultant for GlaxoSmithKline, Insitro and Overtone Therapeutics, and receives research support from Microsoft. D.P. provides consulting service to Ionis Pharmaceuticals, Acadia Pharmaceuticals and M2DS Therapeutics. S.J.S. receives research funding from BioMarin Pharmaceutical. A.O'D.-L. is on the scientific advisory board for Congenica, was a paid consultant for Tome Biosciences, Ono Pharma USA and at present for Addition Therapeutics, and received reagents from PacBio to support rare disease research. Y.C. has a PhD studentship funded by Novo Nordisk. R.A.G. has equity in Codified Genomics. None of these relationships are related to the presented research. The other authors declare no competing interests.

## Additional information

**Extended data** is available for this paper at https://doi.org/10.1038/s41588-026-02554-6.

**Correspondence and requests for materials** should be addressed to Cas Simons or Nicola Whiffin.

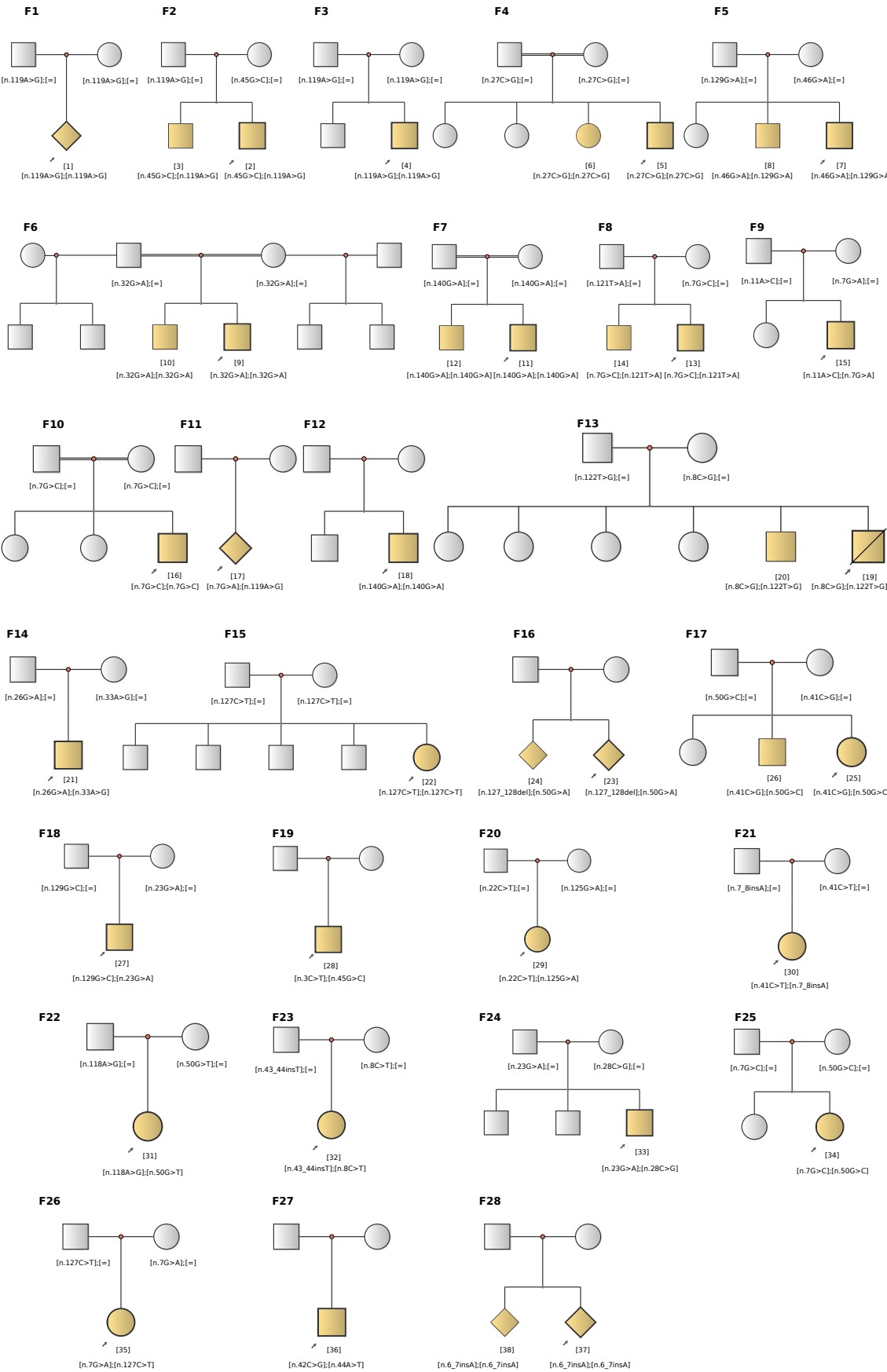

**Extended Data Fig. 1 | Pedigrees and genotyping for all families in the cohort.** Pedigrees are shown for all 28 families. The 38 individuals with NDD are shown in yellow and those without NDD in grey. Genotypes are displayed beneath each individual where available, including parents and unaffected siblings. Individuals without available genotype data have no genotype annotation.

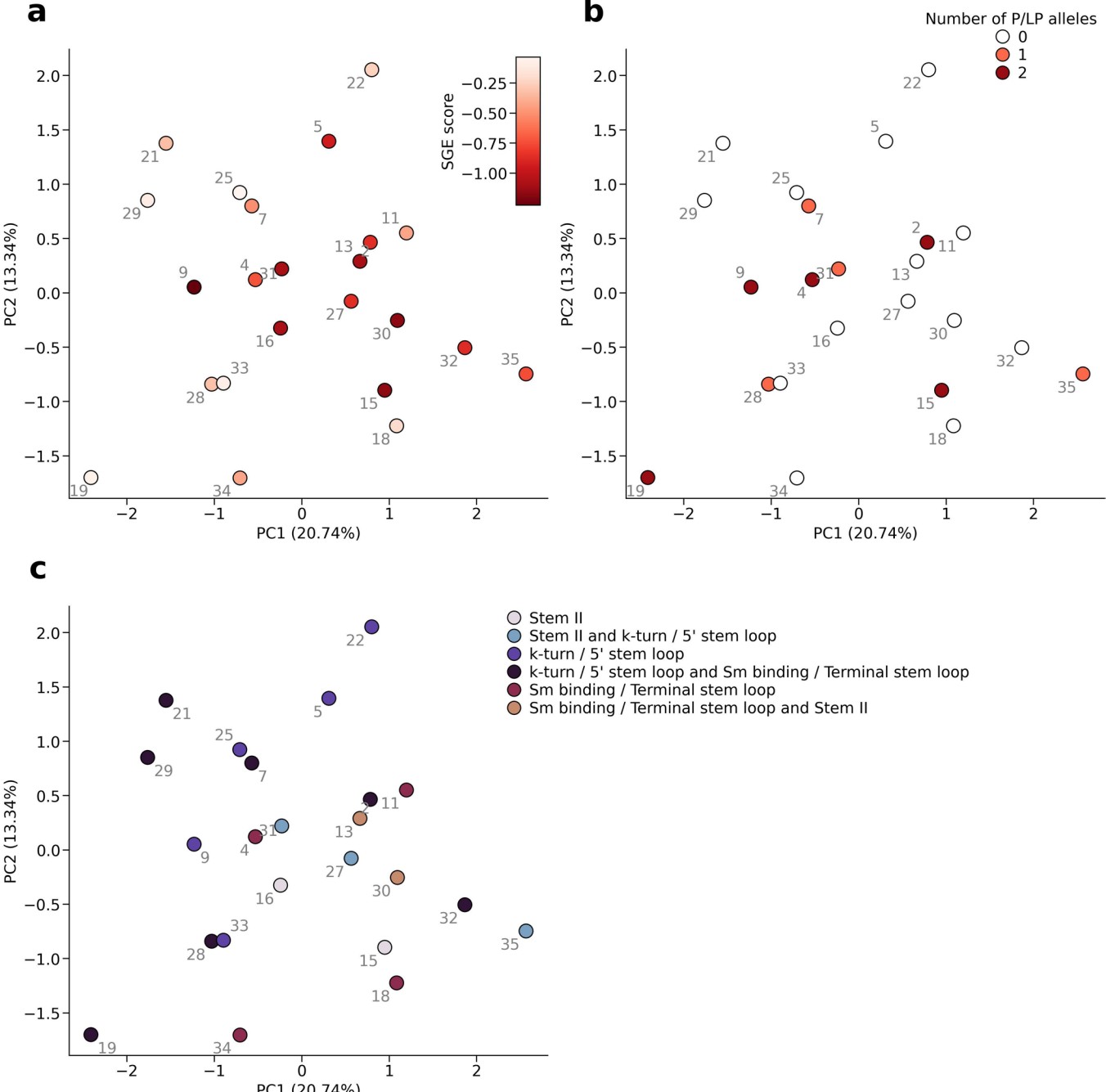

**Extended Data Fig. 2 | Principal components analysis for HPO terms in biallelic *RNU4-2* cases. a**, Points are colored by the SGE score of the most deleterious variant in each individual. **b**, Points are colored by the number of alleles with a Pathogenic / Likely Pathogenic classification. **c**, Points are colored by the position of each biallelic variant pair within the U4 secondary structure. The participant number is shown next to each marker in grey text. Only one individual from each sibling pair is shown. The percentage of the variance explained by each principal component is shown in the axis labels.

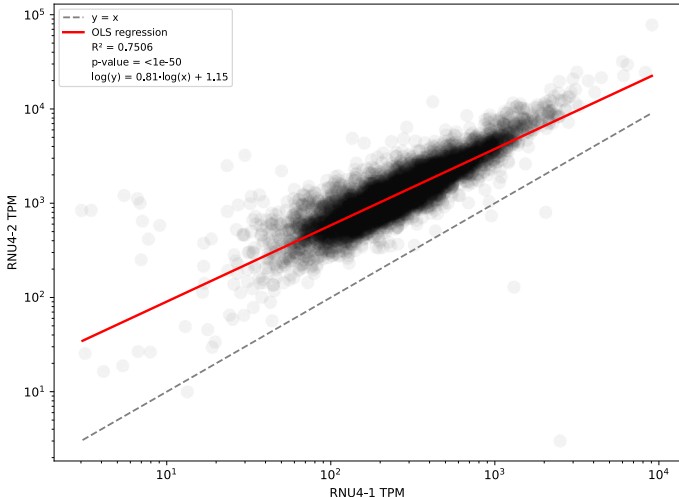

**Extended Data Fig. 3 | Correlation of *RNU4-2* and *RNU4-1* expression across 7,837 RNA-seq samples from GEL.** The red line shows the ordinary least squares regression line of the $\log_{10}$-transformed values. The dashed line shows $y = x$. TPM, Transcripts Per Million.

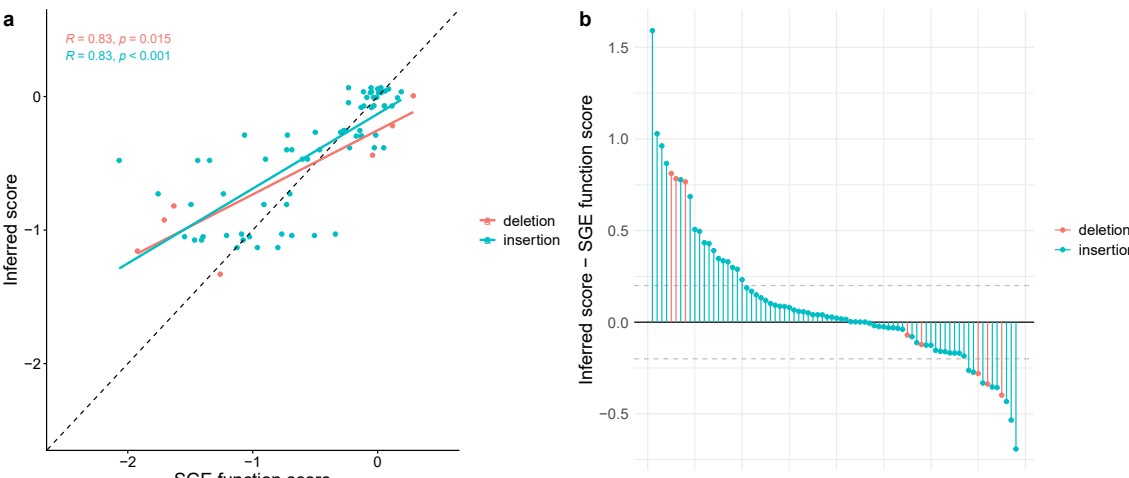

**Extended Data Fig. 4 | Validation of the approach for inferring SGE scores for insertions and deletions not included in the SGE assay. a**, Correlation of the experimentally determined SGE function score to the mean of scores for all SNVs in the flanking (for insertions) or deleted nucleotides (the 'inferred score') for 70 insertions (blue) and eight deletions (red). *R* and *P*-values are from a Spearman's rank test. Exact *P*-value for insertions = $1.24 \times 10^{-18}$. **b**, The difference in inferred and experimentally measured function scores for all 78 tested single-base insertions and deletions. Grey dotted lines correspond to a difference of 0.2.

# Reporting Summary

## Statistics

For all statistical analyses, confirm that the following items are present in the figure legend, table legend, main text, or Methods section.

| n/a | Confirmed | |
|---|---|---|
| ☐ | ☒ | The exact sample size (*n*) for each experimental group/condition, given as a discrete number and unit of measurement |
| ☒ | ☐ | A statement on whether measurements were taken from distinct samples or whether the same sample was measured repeatedly |
| ☐ | ☒ | The statistical test(s) used AND whether they are one- or two-sided<br>*Only common tests should be described solely by name; describe more complex techniques in the Methods section.* |
| ☐ | ☒ | A description of all covariates tested |
| ☐ | ☒ | A description of any assumptions or corrections, such as tests of normality and adjustment for multiple comparisons |
| ☐ | ☒ | A full description of the statistical parameters including central tendency (e.g. means) or other basic estimates (e.g. regression coefficient) AND variation (e.g. standard deviation) or associated estimates of uncertainty (e.g. confidence intervals) |
| ☐ | ☒ | For null hypothesis testing, the test statistic (e.g. *F*, *t*, *r*) with confidence intervals, effect sizes, degrees of freedom and *P* value noted<br>*Give P values as exact values whenever suitable.* |
| ☒ | ☐ | For Bayesian analysis, information on the choice of priors and Markov chain Monte Carlo settings |
| ☒ | ☐ | For hierarchical and complex designs, identification of the appropriate level for tests and full reporting of outcomes |
| ☐ | ☒ | Estimates of effect sizes (e.g. Cohen's *d*, Pearson's *r*), indicating how they were calculated |

*Our web collection on statistics for biologists contains articles on many of the points above.*

## Software and code

Policy information about availability of computer code

| | |
|---|---|
| Data collection | Clinical data were collated using Google sheets and Microsoft Excel spreadsheets. |
| Data analysis | This study predominantly used available software packages: bedtools (v2.31.0), bcftools (v1.16), and samtools (v1.9). R (v4.1.1) was used via RStudio with plots generated using ggplot (v3.5.2) and related packages. Code generated for analyses within this manuscript is stored securely in the National Genomics Research Library and available to be shared within this environment upon request. |

For manuscripts utilizing custom algorithms or software that are central to the research but not yet described in published literature, software must be made available to editors and reviewers. We strongly encourage code deposition in a community repository (e.g. GitHub). See the Nature Portfolio guidelines for submitting code & software for further information.

## Data

Policy information about availability of data

All manuscripts must include a data availability statement. This statement should provide the following information, where applicable:
- Accession codes, unique identifiers, or web links for publicly available datasets
- A description of any restrictions on data availability
- For clinical datasets or third party data, please ensure that the statement adheres to our policy

Data from the National Genomic Research Library (NGRL) used in this research are available within the secure Genomics England Research Environment. Access to NGRL data is restricted to adhere to consent requirements and protect participant privacy. Access to NGRL data is provided to approved researchers who are

members of the Genomics England Research Network, subject to institutional access agreements and research project approval under participant-led governance. For more information on data access, visit: https://www.genomicsengland.co.uk/research

Genomic and phenotypic data from the GREGoR consortium (including the RGP cohort) and the UDN are available through the dbGaP accession numbers phs003047.v1.p1 and phs001232.v5.p2, respectively, with at least annual data releases. Access is managed by a data access committee designated by dbGaP and is based on intended use of the requester and allowed use of the data submitter as defined by consent codes.

Data access to individual genome data from the PFMG2025 with other researchers is submitted to current data protection and regulations in France and is only possible through the Collecteur Analyseur de Données (CAD). More information on data access and the CAD structure can be obtained on the PFMG2025 website (https://pfmg2025.fr/le-plan/collecteur-analyseur-de-donnees-cad/).

UK Biobank data are available in the Research Analysis Platform (UKB-RAP) for use by eligible researchers under approved access. This research was conducted under application number 81050.

RNA-sequencing data in Figure 6A is available in EGA (accession: EGAS50000000889). Data from Figures B-D is available within the NGRL (see access details above).

Variant curations have been deposited in ClinVar under submission SUB16002064 (accessions SCV007494689 - SCV007494723).

All other data are available within the manuscript and supplementary tables.

# Research involving human participants, their data, or biological material

Policy information about studies with human participants or human data. See also policy information about sex, gender (identity/presentation), and sexual orientation and race, ethnicity and racism.

| Reporting on sex and gender | We use the term sex to describe individuals. |
|---|---|
| Reporting on race, ethnicity, or other socially relevant groupings | We record reported ancestry of a subset of participants with detailed clinical information. These data are not used in any analyses. |
| Population characteristics | 31 individuals with biallelic RNU4-2 variants had detailed phenotype characterisation. 21 of these individuals were male, with an average age of 10. |
| Recruitment | Participants were recruited to large genomic sequencing cohorts (e.g. Genomics England) based on clinical presentation. There could be biases from accessibility to recruitment centres. |
| Ethics oversight | Informed consent was obtained for all patients included in this study from their parent(s) or legal guardian, with the study approved by the local regulatory authority. A specific consent form was obtained from the families who consented to the publication of photographs. The 100,000 Genomes Project Protocol has ethical approval from the HRA Committee East of England Cambridge South (REC Ref 14/EE/1112). This study was registered with Genomics England under Research Registry Projects 354. Health related research in UK Biobank was approved by the Research Ethics Committee (REC) under reference 16/NW/0274 with this research conducted under application number 81050. |

Note that full information on the approval of the study protocol must also be provided in the manuscript.

# Field-specific reporting

Please select the one below that is the best fit for your research. If you are not sure, read the appropriate sections before making your selection.

☒ Life sciences ☐ Behavioural & social sciences ☐ Ecological, evolutionary & environmental sciences

For a reference copy of the document with all sections, see nature.com/documents/nr-reporting-summary-flat.pdf

# Life sciences study design

All studies must disclose on these points even when the disclosure is negative.

| Sample size | Sample size was not predetermined, but was based on prior recruitment to genome sequencing cohorts. All undiagnosed individuals with neurodevelopmental disorders were included. |
|---|---|
| Data exclusions | No data were excluded from the analyses. |
| Replication | Replication was achieved by identifying individuals across multiple rare disease cohorts. No other form of replication is relevant to this study as no experiments were performed. |
| Randomization | This is an observational study so randomisation is not relevant. |
| Blinding | This is an observational study so blinding is not relevant. |

# Reporting for specific materials, systems and methods

We require information from authors about some types of materials, experimental systems and methods used in many studies. Here, indicate whether each material, system or method listed is relevant to your study. If you are not sure if a list item applies to your research, read the appropriate section before selecting a response.

## Materials & experimental systems

| n/a | Involved in the study |
|-----|----------------------|
| ☒ | Antibodies |
| ☒ | Eukaryotic cell lines |
| ☒ | Palaeontology and archaeology |
| ☒ | Animals and other organisms |
| ☒ | Clinical data |
| ☒ | Dual use research of concern |
| ☒ | Plants |

## Methods

| n/a | Involved in the study |
|-----|----------------------|
| ☒ | ChIP-seq |
| ☒ | Flow cytometry |
| ☒ | MRI-based neuroimaging |

## Plants

| Seed stocks | NA |
|-------------|-----|
| Novel plant genotypes | NA |
| Authentication | NA |

