## [Peer Review File · Nature Genetics]

Biallelic variants in the non-coding RNA gene RNU4-2 cause a recessive neurodevelopmental syndrome with distinct white matter changes

Corresponding Author: Dr Nicola Whiffin

Version 0:

Decision Letter:

24th October 2025

Dear Nicky,

Your Article "Biallelic variants in the non-coding RNA gene RNU4-2 cause a recessive neurodevelopmental syndrome with distinct white matter changes" has been seen by three referees. You will see from their comments below that, while they find your work of interest, they have raised overlapping concerns. We are interested in the possibility of publishing your study in Nature Genetics, but we would like to consider your response to these concerns in the form of a revised manuscript before we make a final decision on publication.

To guide the scope of the revisions, the editors discuss the referee reports in detail within the team, including with the chief editor, with a view to identifying key priorities that should be addressed in revision, and sometimes overruling referee requests that are deemed beyond the scope of the current study. In this case, we particularly ask that you clarify the overlap and distinguishing features of this study compared to the related submission reporting individuals with this recessive RNU4-2 syndrome, provide more details on clinical phenotypes in comparison to the dominant RNU4-2 syndrome, and extend the mechanistic studies as requested by the reviewers. We hope you will find this prioritized set of referee points to be useful when revising your study. Please do not hesitate to get in touch if you would like to discuss these issues further.

We therefore invite you to revise your manuscript taking into account all reviewer and editor comments. Please highlight all changes in the manuscript text file. At this stage, we will need you to upload a copy of the manuscript in MS Word .docx or similar editable format.

*2) If you have not done so already, please begin to revise your manuscript so that it conforms to our Article format instructions, available

http://www.nature.com/ng/authors/article_types/index.html here

*3) Include a revised version of your Reporting Summary: <https://www.nature.com/documents/nr-reporting-summary.pdf> It will be available to referees (and, potentially, statisticians) to aid in their evaluation if the manuscript goes back for peer review.

*4) Include the most recent version of the saturated genome editing (SGE) study as a related manuscript file so it will be able to the editors and reviewers for comparative assessment.

Please be aware of our [guidelines](https://www.nature.com/nature-research/editorial-policies/image-integrity) on digital image standards.

EXTENDED DATA FIGURES

Link Redacted

We hope to receive your revised manuscript within 4-8 weeks. If you cannot send it within this time, please let us know.

Nature Genetics is committed to improving transparency in authorship. As part of our efforts in this direction, we are now requesting that all authors identified as 'corresponding author' on published papers create and link their Open Researcher and Contributor Identifier (ORCID) with their account on the Manuscript Tracking System (MTS), prior to acceptance. ORCID helps the scientific community achieve unambiguous attribution of all scholarly contributions. You can create and link your ORCID from the home page of the MTS by clicking on 'Modify my Springer Nature account'. For more information, please visit www.springernature.com/orcid.

Sincerely,
Kyle

Kyle Vogan, PhD
Senior Editor
Nature Genetics
<https://orcid.org/0000-0001-9565-9665>

Referee expertise:

Referee #1: Genetics, rare diseases, developmental disorders

Referee #2: Genetics, rare diseases, developmental disorders

Referee #3: Genetics, rare diseases, developmental disorders

Reviewers' Comments:

Reviewer #1 (Remarks to the Author):

* Summary of the key results

Description of a new molecular cause in a fraction of cases with neurodevelopmental delay (NDD): a recessive phenotype due to likely pathogenic variants in the RNU4-2 snRNA gene.

* Originality and significance: if not novel, please include reference

I have the following problem with this manuscript: The authors state "In a companion paper (reference 13 deJonghe et al, a medRxiv preprint that - my information is - has been submitted for publication to a high impact Journal), we describe a saturation genome editing functional experiment, where we simultaneously measured the functional impact of variants across RNU4-2 in a haploid cell line. These data led us to identify a novel recessive NDD caused by biallelic variants outside of the T-loop and Stem III regions in which heterozygous variants cause ReNU syndrome". In the medRxiv preprint of ref 13, there is a description of 19 cases with recessive RNU4-2 homozygous or compound heterozygous variants in RNU4-2 causing NDD. These variants are in regions of the gene distinct from the dominant RNU4-2 variants. Thus, formally, the recessive syndrome has already been described. The present Rius et al manuscript describes 32 cases with recessive RNU4-2 variants (the overlap with the deJonghe et al paper is not clear) as a follow up of the deJonghe et al paper. As the

authors of this manuscript state "Here, we describe the clinical phenotype of this novel autosomal recessive NDD caused by variants in RNU4-2". I guess the "original" contribution of this manuscript is the clinical description of the phenotype and its comparison with that of the dominant RNU4-2-related NDD.

* Data & methodology: validity of approach, quality of data, quality of presentation
High quality of data and excellent presentation

* Appropriate use of statistics and treatment of uncertainties
Yes

* Conclusions: robustness, validity, reliability
The conclusion that "Dominant and recessive RNU4-2 NDDs have distinct phenotypic features" could be also seen (based on Figure 2B) as an overlap or continuum of phenotypes with some more frequent in the dominant and others as more frequent in the recessive form. Please see the analogous conclusions of the RNU2-2 medRxiv paper on dominant and recessive variants causing NDD by Leitao et al <https://pubmed.ncbi.nlm.nih.gov/40950445/>.

* Suggested improvements: experiments, data for possible revision

* References: appropriate credit to previous work?
Yes

* Clarity and context: lucidity of abstract/summary, appropriateness of abstract, introduction and conclusions
Yes

Reviewer #2 (Remarks to the Author):

A. Summary of key results: This study is a companion to a manuscript recently reviewed in Nature by De Jonghe and colleagues, which leverages saturated genome editing (SGE) to systematically identify the functional impacts of variants across the RNU4-2 locus. RNU4-2 has garnered substantial attention in recent years because de novo pathogenic variants in this ncRNA are thought to be the most prevalent cause of neurodevelopmental disorders (NDD) in a clinical entity termed ReNU syndrome. SGE was pursued to experimentally determine which variants within RNU4-2 are pathogenic/likely pathogenic and to determine whether there is a correlation between phenotype and region of the ncRNA that is impacted. Fortuitously, SGE data led the authors to identify variants outside of the T-loop and Stem III regions in which heterozygous variants cause ReNU syndrome. Here, Rius et al. expand the contribution of non-coding small nuclear RNAs to NDD by reporting 32 individuals who harbor biallelic variants in RNU4-2. Additionally, they present comprehensive clinical phenotyping of the cohort and suggest that the recessive disorder is not only genetically distinct, but also clinically distinct from ReNU syndrome.

B. Originality and significance: This study is impactful in its elucidation of the disease outcomes stemming from variants in RNU4-2 that are divergent from a clinical and inheritance standpoint.

C. Data & Methodology: The study is grounded on rich genetic data from large global NDD cohorts, has beautifully reported clinical features (Figures 2 and 4 and Table 1), includes brain imaging data from a substantial fraction of the cohort (24 of 32 individuals; Figure 3), is supported by SGE function scores (Figures 1 and 5), and offers careful variant curation according to the ACMG/AMP framework.

D. Appropriate use of statistics: No concerns.

E. Conclusions: The conclusions are supported by an ample cohort size, robust enrichment studies, and appropriate caveats with the SGE predictions.

F. To meet the rigorous standards of Nature Genetics, I have suggestions to improve the manuscript further.

Major points:

1. I also reviewed the SGE manuscript by De Jonghe and will reiterate my major concern for the pair of manuscripts on RNU4-2: neither manuscript answers the question of why the recessive and dominant forms are caused by variants in different regions of the ncRNA, and how those variants might contribute to distinct phenotypes. This functional basis (or at least a starting point) is much needed to enable eventual therapeutic development, and I argue that generation of such functional data is not beyond the scope of the current work. At minimum, RNAseq should be performed on cell lines from affected individuals with the recessive RNU4-2 NDD for comparison with the ReNU samples reported by Chen et al. 2024 (Nature). Even if access to samples from affected individuals is not possible, the authors have demonstrated their expertise with CRISPR/Cas9 and could edit these regions of the gene to simulate biallelic variants so that global impacts on splicing can be determined.

2. If merged with the SGE paper, the entire body of work would elevate the field substantially. However, as a stand-alone manuscript, this paper does not advance the field sufficiently to warrant publication in Nature Genetics. In its current form, this paper reports a new rare NDD entity (clinical spectrum and inheritance pattern) from a ncRNA already implicated in disease but does not offer a biological explanation of why.

3. The statements about divergent clinical features between recessive and dominant forms are contradictory, likely because of a lack of detailed phenotyping and/or lack of sufficient numbers of individuals. Phenotype enrichment analysis in Figure 2B suggests that there is no significant enrichment of phenotypic features between the two disorders when the 27 recessive individuals are compared to 178 individuals with dominant ReNU syndrome (and after FDR correction). Thus, if the authors want to distinguish the two conditions, more emphasis should go toward detailing the purportedly different aspects. Two suggestions might be the following: (a) Given that brain MRI findings are reported here to be a distinguishing feature between ReNU syndrome and the recessive RNU4-2 NDD, the authors should include more details in the form of a dedicated supplementary table (e.g. in the same style as Fig 2A) to include age at time of imaging, and precise abnormalities found on the individual level (white matter changes, perivascular spaces, thin corpus callosum, cerebellum). Although the summary in Table 1 and Supplementary Table 4 text are informative, individual level data broken down by features will more easily reveal the most common combinations of MRI findings and progression. (b) If facial features are truly different in dominant vs recessive forms of RNU4-2 NDD, the authors might consider facial image analysis between individuals from each group to show that the differences are significant, and not just anecdotal as described in the main text and shown in Figure 3. Addressing this point will be crucial toward concluding that these are actually two distinct disorders rather than a continuum of the same condition caused by dysfunctional RNU4-2.

Minor points

1. There are multiple affected siblings reported in the manuscript. Can the authors comment on phenotype concordance or divergent phenotypes among sibs? Please clarify in Supplementary Table 4 which individuals are siblings. A column with a family identifier could help.
 2. Although I would like to trust the aggregate genetic investigation of the collective team, the manuscript would benefit from a supplementary figure which includes pedigrees from all 32 individuals to show readers that due diligence was taken to genotype parents and unaffected siblings. If genotype data are missing from family members, this would be an opportunity to show this on the individual level.
 3. Supplementary Table 1. Clarify the significance of the color-coded cells in the table. Do the different colors indicate recurrent variants?
 4. Please include the human genome reference build in Supplementary Tables 1, 2, 3.
 5. Supplementary Table 4. Please avoid blank cells, and especially for phenotypic features, use NA (not available) or N (not present) to complete the entire table.
 6. Figure 1. SGE scores for insertions and deletions without available SGE data were inferred from the mean SGE score across all SNVs within the deleted nucleotides or taking the mean SGE score across all SNVs. Is there a published precedent for this, or are there experimental data that indicate that such an inference approach will not introduce false positive results?
- G. References: Citations are appropriate and current.
- H. Clarity and context: The organization and writing style of the manuscript is clear.

Reviewer #3 (Remarks to the Author):

In this manuscript, authors report a cohort of individuals with a neurodevelopmental disorder associated with biallelic variants in RNU4-2. De novo dominant-acting variants in the same gene have recently been described as causing the prevalent neurodevelopmental disorder ReNU syndrome. The authors demonstrate that biallelic variants associated with NDD are found outside of the ReNU syndrome region of the gene, but clustered within regions which encode functionally important elements of U4. Biallelic variants in RNU4-2 are rare in healthy population cohorts, and enriched in cohorts with neurodevelopmental disorders. A companion manuscript performs saturation genome editing of RNU4-2, and individuals with NDD tend to have variants with a lower SGE score than individuals in the UKBB with biallelic RNU4-2 variants. This is an important study that has been carefully conducted. Statistics are robust and appropriate. It is likely to be of interest to a broad readership including the neurology and genetics clinical communities, as well as those interested in the spliceosome and RNA biology. From a clinical perspective, awareness of both dominant and recessive conditions caused by variants in the same gene is crucial, in order to appropriately counsel families about the chance of future children in the family be affected.

Major comments:

1. Impact on splicing

Given the reduced sensitivity of the SGE assay for the biallelic variants, and the high rate of mutation rate in these genes, additional mechanistic information on the impact of these variants on splicing will be invaluable and will be of immediate value in clinical practice. Chen et al. previously characterised the impact on splicing by performing RNAseq on ReNU patient samples. A similar experiment should be performed on cell lines or blood samples from affected individuals with the recessive RNU4-2 NDD. A common clinical scenario in the future will be interpreting RNU4-2 variant/s in patients with a compatible but nonspecific phenotype, such as global developmental delay or intellectual disability, and as the SGE scores

lack sensitivity for the recessive condition, such data will be invaluable. If access to patient-derived samples is not possible, the authors could use CRISPR/Cas9 to generate cellular models of the dominant and recessive conditions for splicing analysis.

2. Neuroradiological phenotypes

The phenotypes of individuals affected by dominant and recessive conditions seem strikingly similar, with the presence of periventricular perivascular dilation and cerebellar atrophy important distinguishing features of the recessive condition. Were MRI scans of individuals with ReNU syndrome specifically reviewed to look for signs of perivascular dilation and/or cerebellar atrophy? How does the perivascular dilatation reported here differ from the relatively common finding of cystic periventricular leukomalacia? Is this easy or difficult to distinguish? I note that the initial MRI reports for several individuals (presented in supplementary table 4) describe these anomalies as cystic periventricular leukomalacia. I feel this is important because a finding of cystic periventricular leukomalacia may be interpreted by clinicians as indicating an antenatal or perinatal insult (commonly seen in prematurity for example) and may reduce the likelihood of genetic testing being offered, particularly if the birth history is compatible. Secondly, some more detail on the phenotype of the cerebellar atrophy would be welcome – is this generalised, or affecting a particular region of the cerebellum?

3. UKBB phenotypes

Can you comment on the phenotypes of the individuals in UKBB who are compound heterozygous for RNU4-2 variants? What about the individual who has variants with significant SGE scores? Do you think that this indicates variable expressivity or penetrance?

4. SGE scores

The companion manuscript designates an SGE score < -0.39 as significant. However, this manuscript describes an SGE score of -0.2 as being the threshold that best distinguishes variants carried by individuals with a neurodevelopmental disorder from UKBB. However, both thresholds are used in this publication, which introduces confusion, as to which is most appropriate to use clinically. I appreciate that the authors are reluctant to be too prescriptive, but a bit more clarity would be welcome – do they recommend the use of an average SGE score, or a threshold of -0.2 or -0.39 in evaluating biallelic variants, do they recommend that this score differs based on the region that the variants lie within U4 (i.e., for putative ReNU syndrome de novo variants vs biallelic variants)?

Minor comments:

1. Table 1: Are the cerebellar anomalies being discussed cerebellar atrophy? In which case it would be useful to name this directly. Alternatively, if there are other anomalies observed, please give details.

2. Figure 5B:

a. A significant SGE score is given as function score < -0.39 , but earlier (in the comparison to UKBB variants) a threshold of -0.2 was applied. Would it not be more appropriate to use -0.2 ?

b. It would be helpful to continue the shading colours from 5A to describe the functional domains. For example, could the ReNU syndrome critical regions be shaded in light blue/green, rather than darker gray?

c. E it would be useful to plot the homozygous individuals in UKBB as solid circles together with the heterozygotes in D, or immediately below them, rather than with the UKBB allele frequency information, I found it hard to work out exactly where these variants are situated with respect to the SGE heatmap and functional domains (I think the position of the variant is indicated by a grey bar behind the UKBB allele frequency plot, but I'm not fully certain).

d. I presume in 5E colour reflects UKBB allele frequency? It would be useful to have the parameters of this in the key.

e. It would be useful to colour the UKBB individuals' variants in D and E in the same way that the NDD individuals' variants are coloured in B, to make clear which are the variants that have a significant SGE score.

Supplementary Table 4

1. This is a useful table for a clinician to get a feeling for these patients and how they present, thank you for including this table and the detailed information.

2. However, it is quite hard to read the table in the current format because there are no row-lines, and information runs over several pages, often without headers, making it hard to work out which individual some of the features pertain to. My apologies if this is to do with how I opened the file. This would be much better as an excel spreadsheet. Alternatively, please orientate in a landscape direction, and provide the individual's number as the first column on each page, and the column headings on each page.

3. Why does it start with individual 2, not 1, is this information missing, or is this a numbering error?

4. There are multiple spelling mistakes, a spell-check would be useful.

5. I don't understand what 'false routes' in the other phenotypes section means: e.g., 'with false routes and apnoeas during

crying' and 'stridor associated with false routes'. Does this mean trachea-oesophageal fistula? Does it mean aspiration of liquid into the lungs? Does it mean coughing/stridor after feeds/with secretions?

6. Please report anthropometric measurements such as birth weight and heights in a standard way, e.g., the measure plus the gestational age corrected centile. For some measurements no centiles are given, for others Z-scores are given, for others a notation such as [3.P] is given, which I'm afraid I don't understand the significance of. Consistency would be useful, I would recommend centiles.

7. Consistency of reporting of EEG features would be useful, e.g., for each EEG comment on:

- a. is the background normal/abnormal?
- b. Were spike-wave discharges seen; if so, are they focal or generalised; if focal where?
- c. Were epileptic seizures seen; were they focal or generalised; if focal, where?
- d. Any other specific features suggestive of a particular epilepsy syndrome.

One entry refers to 'epilepsy predisposition', what does this mean (i.e., it would be more useful to give the observations that were made on the EEG that led the neurophysiologist to this conclusion)?

8. It would be useful, if possible, to know whether epilepsy was controlled with medication or was drug resistant/required multiple anti-epileptic medications.

Version 1:

Decision Letter:

Our ref: NG-A70065R

10th February 2026

Dear Nicky,

Your revised manuscript "Biallelic variants in the non-coding RNA gene RNU4-2 cause a recessive neurodevelopmental syndrome with distinct white matter changes" (NG-A70065R) has been seen by the original referees. As you will see from their comments below, they find that the paper has improved in revision, and therefore we will be happy in principle to publish it in Nature Genetics as an Article pending final revisions to satisfy their remaining requests and to comply with our editorial and formatting guidelines.

We are now performing detailed checks on your paper, and we will send you a checklist detailing our editorial and formatting requirements soon. Please do not upload the final materials or make any revisions until you receive this additional information from us.

Thank you again for your interest in Nature Genetics. Please do not hesitate to contact me if you have any questions.

Sincerely,
Kyle

Kyle Vogan, PhD
Senior Editor
Nature Genetics
<https://orcid.org/0000-0001-9565-9665>

Reviewer #1 (Remarks to the Author):

The revised document answers all the comments of the reviewers and provides important additional RNA-seq data that further characterize the differences between the dominant and recessive RNU4-2 diseases.

Reviewer #2 (Remarks to the Author):

A. Summary

This article and its companion paper under review in Nature have been substantially elevated by new data, and both are markedly improved. The main changes to the manuscript are as follows: (1) addition of analysis of RNAseq data from patients with biallelic RNU4-2 variants; (2) they support claims of a phenotype distinction between dominant and recessive

disease caused by pathogenic RNU4-2 variants (with emphasis on the facial gestalt and MRI findings); and (3) add five additional affected individuals to the case series.

B. Originality

Rius et al. expand the contribution of non-coding small nuclear RNAs to NDD by reporting 38 individuals who harbor biallelic variants in RNU4-2. Additionally, they present comprehensive clinical phenotyping of the cohort and suggest that the recessive disorder is genetically, clinically, and molecularly distinct from ReNU syndrome.

C. Data and methodology

The data are robust and the presentation is clear.

D. Statistics

No concerns.

E. Conclusions

The study is grounded on rich genetic data from large global NDD cohorts, has beautifully (and even more improved) reported clinical features, includes brain imaging data from a substantial fraction of the cohort, is supported by SGE function scores from the companion paper (in review at Nature), and offers careful variant curation according to the ACMG/AMP framework. This study is impactful in its elucidation of the disease outcomes stemming from variants in RNU4-2 that are divergent from the recessive NDD.

F. Suggested improvements

I am grateful to the authors for their attentiveness to my previous major concerns about the manuscript and their new data address my comments to satisfaction. I would be delighted to accept this important body of work in Nature Genetics with the provision that the two minor comments be addressed:

1. Line 354, Results: The authors state that phenotype concordance was high across the seven multiplex families, however variable expressivity was observed in 3 families. The latter argues against high concordance, so the statement of "high concordance" should be softened.
2. Line 454, Results: Correct the p-values to include superscript.

G. References

No concerns; citations are appropriate.

H. Clarity and context

The work is appropriately placed in the context of RNA splicing defects in disease and has an appropriate abstract, introduction and discussion.

Reviewer #3 (Remarks to the Author):

In this manuscript, authors report a cohort of individuals with a neurodevelopmental disorder associated with biallelic variants in RNU4-2. De novo dominant-acting variants in the same gene have recently been described as causing the prevalent neurodevelopmental disorder ReNU syndrome. The authors demonstrate that biallelic variants associated with NDD are found outside of the ReNU syndrome region of the gene, but clustered within regions which encode functionally important elements of U4. Biallelic variants in RNU4-2 are rare in healthy population cohorts, and enriched in cohorts with neurodevelopmental disorders. A companion manuscript performs saturation genome editing of RNU4-2, and individuals with NDD tend to have variants with a lower SGE score than individuals in the UKBB with biallelic RNU4-2 variants. In this revised manuscript, the authors include RNA-sequencing data of individuals with de novo and biallelic RNU4-2 variants, providing a valuable potential biomarker to identify the recessive condition.

Major comments:

1. Impact on splicing

The authors include RNA-sequencing data from patients with both dominant and recessive RNU4-2 variants, providing a useful biomarker to distinguish these conditions.

2. Neuroradiological phenotypes

Thank you for providing more detail regarding the neuroradiological phenotypes. The additional detail regarding ReNU syndrome phenotypes is welcome. This suggests that the perivascular space dilatation may also be seen in ReNU syndrome (although less prominently), as reflected by Figure 2B.

3. SGE scores

The clarifications provided by the authors are helpful, thank you. I presume that the authors' suggest that clinical labs should apply an evidence strength of supporting when variants have statistically significant SGE scores (as they themselves have done when classifying variants according to the ACMG guidelines), until an independently ascertained cohort is available to properly calibrate the scores? It would be helpful to add a sentence to this effect to the relevant section of the discussion.

The authors have adequately addressed all of my concerns and I would recommend the paper be accepted for publication.

E Radford

We thank the reviewers for their valuable comments on our manuscript. We have made considerable changes to the manuscript in response, which are detailed below. Additions/edits to the text are shown below and in the manuscript as **pink text**.

Summary of significant changes

The most significant change we have made is the addition of analysis of RNA-sequencing data from patients with biallelic *RNU4-2* variants. Using these data we show that:

- (1) Individuals with biallelic *RNU4-2* variants do not share the 'signature' of 5' splice site disruption that is observed in individuals with ReNU syndrome (**Fig. 6A**)
- (2) Biallelic *RNU4-2* variants are associated with a dramatic loss of *RNU4-2* expression and a corresponding elevation in levels of the paralogous gene *RNU4-1*. This is in contrast to individuals with ReNU syndrome who have marginally elevated *RNU4-2* levels. These data are consistent with different molecular mechanisms underlying the recessive and dominant NDDs that could explain differences in observed phenotypes (**Fig. 6B, C**).
- (3) The ratio of *RNU4-2* to *RNU4-1* expression levels are a potential diagnostic biomarker for the recessive condition.
- (4) We do not observe a reproducible splicing defect in blood for the recessive *RNU4-2* disorder (**Fig. 6D**). We suggest that any splicing disruption would be very subtle. These data are consistent with similar analyses for a recently discovered recessive disorder in *RNU2-2*.

We have included these new results in a separate section of the results and in a new Figure 6. We have also added text to the abstract, methods, and discussion sections around these findings.

Further, we have added additional data and analyses to support our claims of a phenotypic distinction between the recessive and dominant disorders:

- (1) An analysis using GestaltMatcher to show that individuals with the recessive *RNU4-2* NDD have a convergent facial phenotype that is not shared with the ReNU syndrome facial gestalt. Individuals with ReNU syndrome are no more similar to individuals with biallelic *RNU4-2* variants than to random individuals with other disorders (**Fig. 3B**).
- (2) Dedicated Supplementary Tables to detail the MRI findings in 27 individuals with biallelic *RNU4-2* including thirteen scans reviewed by a single paediatric neuroradiologist, along with nine MRI scans from individuals with ReNU for direct comparison.

We have made changes to this manuscript to align with our response to the reviewers of the companion manuscript. In particular, we have adjusted the SGE scores and related score thresholds after performing a third replicate of the experiment. These adjusted scores do not change any of the results significantly, or any of the associated conclusions.

Finally, we have added five additional patients to our cohort that were identified after our initial submission. This brings our total cohort to 38 patients from 28 families. We have detailed clinical information for 31 of these patients. We have changed the number identifiers for the identified individuals such that the 20 included in the companion manuscript are numbered 1-20, the additional individuals included in our characterised cohort are numbered 21-38, and the excluded individuals (based on mean SGE scores) are numbered 39-43.

Reviewer #1

* Summary of the key results

Description of a new molecular cause in a fraction of cases with neurodevelopmental delay (NDD): a recessive phenotype due to likely pathogenic variants in the RNU4-2 snRNA gene.

* Originality and significance: if not novel, please include reference

I have the following problem with this manuscript: The authors state "In a companion paper (reference 13 deJonghe et al, a medRxiv preprint that - my information is - has been submitted for publication to a high impact Journal), we describe a saturation genome editing functional experiment, where we simultaneously measured the functional impact of variants across RNU4-2 in a haploid cell line. These data led us to identify a novel recessive NDD caused by biallelic variants outside of the T-loop and Stem III regions in which heterozygous variants cause ReNU syndrome". In the medRxiv preprint of ref 13 there is a description of 19 cases with recessive RNU4-2 homozygous or compound heterozygous variants in RNU4-2 causing NDD. These variants are in regions of the gene, distinct from the dominant RNU4-2 variants. Thus, formally, the recessive syndrome has already been described. The present Rius et al manuscript describes 32 cases with recessive RNU4-2 variants (the overlap with the deJonghe et al paper is not clear) as a follow up of the deJonghe et al paper. As the authors of this manuscript state "Here, we describe the clinical phenotype of this novel autosomal recessive NDD caused by variants in RNU4-2". I guess the "original" contribution of this manuscript is the clinical description of the phenotype and its comparison with that of the dominant RNU4-2-related NDD.

Response: We have designed this current manuscript as a companion to the De Jonghe *et al.* paper, which details our route to discovery of the recessive NDD. We disagree that this is a follow-up or that formally the recessive syndrome has already been described: these two papers, in combination, form that initial description. The reason that the De Jonghe *et al.* paper was preprinted first, was solely due to the time it takes to collect detailed clinical phenotype data for a substantial cohort of patients and to carefully curate these data. The De Jonghe *et al.* manuscript does not detail the clinical phenotype of the recessive NDD, or compare it to ReNU syndrome, either phenotypically or mechanistically; those analyses are in this paper. We believe that the extent and importance of these analyses justify this second manuscript.

In this manuscript, we expand the spectrum of the recessive NDD from what was discovered using the SGE, creating a significant cohort for a novel gene-disease relationship. We do appreciate that in doing this, the overlap in patients between the two manuscripts was not clear. We have now added the text to make this overlap more evident in the second paragraph of the results: "In total, we identified 43 individuals across 33 families, including the 20 reported in De Jonghe *et al.* (Sup. Table 1)." We have also added a column to Supplementary Table 1 to detail which patients are included in the companion paper.

We have also increased the new data presented in this manuscript by including the analyses of RNA-seq data that are described above. These data beautifully highlight the difference in

mechanisms underlying the dominant and recessive NDDs adding to the original contribution from this manuscript.

* Data & methodology: validity of approach, quality of data, quality of presentation
High quality of data and excellent presentation

* Appropriate use of statistics and treatment of uncertainties
Yes

* Conclusions: robustness, validity, reliability
The conclusion that "Dominant and recessive RNU4-2 NDDs have distinct phenotypic features" could be also seen (based on Figure 2B) as an overlap or continuum of phenotypes with some more frequent in the dominant and others as more frequent in the recessive form. Please see the analogous conclusions of the RNU2-2 medRxiv paper on dominant and recessive variants causing NDD by Leitao et al
<https://pubmed.ncbi.nlm.nih.gov/40950445/>.

Response: Thank you for bringing up this point. Our presentation as distinct syndromes is driven by our current mechanistic understanding, which is now supported by the RNA-seq results described above, and the distinct MRI phenotype in the recessive cases. Nevertheless, we have now added a section to the discussion to note this alternative viewpoint. This reads: "The dominant and recessive *RNU4-2* associated NDDs have many overlapping phenotypic features, which could indicate a continuum or phenotypic spectrum across the two disorders."

* Suggested improvements: experiments, data for possible revision

* References: appropriate credit to previous work?
Yes

* Clarity and context: lucidity of abstract/summary, appropriateness of abstract, introduction and conclusions
Yes

Reviewer #2

A. Summary of key results: This study is a companion to a manuscript recently reviewed in Nature by De Jonghe and colleagues, which leverages saturated genome editing (SGE) to systematically identify the functional impacts of variants across the RNU4-2 locus. RNU4-2 has garnered substantial attention in recent years because de novo pathogenic variants in this ncRNA are thought to be the most prevalent cause of neurodevelopmental disorders (NDD) in a clinical entity termed ReNU syndrome. SGE was pursued to experimentally determine which variants within RNU4-2 are pathogenic/likely pathogenic and to determine whether there is a correlation between phenotype and region of the ncRNA that is impacted. Fortuitously, SGE data led the authors to identify variants outside of the T-loop and Stem III regions in which heterozygous variants cause ReNU syndrome. Here, Rius et al. expand the contribution of non-coding small nuclear RNAs to NDD by reporting 32 individuals who harbor biallelic variants in RNU4-2. Additionally, they present comprehensive clinical

phenotyping of the cohort and suggest that the recessive disorder is not only genetically distinct, but also clinically distinct from ReNU syndrome.

B. Originality and significance: This study is impactful in its elucidation of the disease outcomes stemming from variants in RNU4-2 that are divergent from a clinical and inheritance standpoint.

C. Data & Methodology: The study is grounded on rich genetic data from large global NDD cohorts, has beautifully reported clinical features (Figures 2 and 4 and Table 1), includes brain imaging data from a substantial fraction of the cohort (24 of 32 individuals; Figure 3), is supported by SGE function scores (Figures 1 and 5), and offers careful variant curation according to the ACMG/AMP framework.

D. Appropriate use of statistics: No concerns.

E. Conclusions: The conclusions are supported by an ample cohort size, robust enrichment studies, and appropriate caveats with the SGE predictions.

F. To meet the rigorous standards of Nature Genetics, I have suggestions to improve the manuscript further.

Major points:

1. I also reviewed the SGE manuscript by De Jonghe and will reiterate my major concern for the pair of manuscripts on RNU4-2: neither manuscript answers the question of why the recessive and dominant forms are caused by variants in different regions of the ncRNA, and how those variants might contribute to distinct phenotypes. This functional basis (or at least a starting point) is much needed to enable eventual therapeutic development, and I argue that generation of such functional data is not beyond the scope of the current work. At minimum, RNAseq should be performed on cell lines from affected individuals with the recessive RNU4-2 NDD for comparison with the ReNU samples reported by Chen *et al.* 2024 (Nature). Even if access to samples from affected individuals is not possible, the authors have demonstrated their expertise with CRISPR/Cas9 and could edit these regions of the gene to simulate biallelic variants so that global impacts on splicing can be determined.

Response: We agree that our initial manuscripts did not adequately address this important question, but at that time we did not have RNA-seq data available to study this. As described in the summary at the top of this document, we have now included analysis of RNA-seq data both from patients with biallelic *RNU4-2* variants and from individuals with ReNU syndrome. We were very excited to see that these data begin to explain why, mechanistically, these different variants result in distinct phenotypes with distinct inheritance patterns. Specifically, these data suggest that the recessive NDD may be a consequence of loss of *RNU4-2* expression. This is in contrast to ReNU syndrome, which our previous analyses (Chen *et al.* Nature 2024; Nava *et al.* Nature Genetics 2025) suggest has a 'poison allele' or 'altered function' mechanism. As mentioned above, we have included these new results in a separate section at the end of the results (lines 539-584) and in a new Figure (**Fig. 6**). We have also added text to the abstract (lines 185-187), methods (lines 846-889), and discussion (lines 629-648) sections around these new findings.

2. If merged with the SGE paper, the entire body of work would elevate the field substantially. However, as a stand-alone manuscript, this paper does not advance the field sufficiently to warrant publication in Nature Genetics. In its current form, this paper reports a new rare NDD entity (clinical spectrum and inheritance pattern) from a ncRNA already implicated in disease but does not offer a biological explanation of why.

Response: This manuscript describes a novel recessive NDD through characterisation of a substantial cohort of patients with detailed clinical information. With addition of the above described RNA-seq analysis, we now compare this recessive NDD to dominant ReNU syndrome at the genetic, phenotypic, and mechanistic levels. Critically, the added RNA-seq data gives that biological explanation of why we are seeing two distinct syndromes. We believe that the extent and importance of these analyses justify this second manuscript and have discussed this with the editors.

3. The statements about divergent clinical features between recessive and dominant forms are contradictory, likely because of a lack of detailed phenotyping and/or lack of sufficient numbers of individuals. Phenotype enrichment analysis in Figure 2B suggests that there is no significant enrichment of phenotypic features between the two disorders when the 27 recessive individuals are compared to 178 individuals with dominant ReNU syndrome (and after FDR correction). Thus, if the authors want to distinguish the two conditions, more emphasis should go toward detailing the purportedly different aspects. Two suggestions might be the following: (a) Given that brain MRI findings are reported here to be a distinguishing feature between ReNU syndrome and the recessive RNU4-2 NDD, the authors should include more details in the form of a dedicated supplementary table (e.g. in the same style as Fig 2A) to include age at time of imaging, and precise abnormalities found on the individual level (white matter changes, perivascular spaces, thin corpus callosum, cerebellum). Although the summary in Table 1 and Supplementary Table 4 text are informative, individual level data broken down by features will more easily reveal the most common combinations of MRI findings and progression. (b) If facial features are truly different in dominant vs recessive forms of RNU4-2 NDD, the authors might consider facial image analysis between individuals from each group to show that the differences are significant, and not just anecdotal as described in the main text and shown in Figure 3. Addressing this point will be crucial toward concluding that these are actually two distinct disorders rather than a continuum of the same condition caused by dysfunctional RNU4-2.

Response: Thank you for these suggestions. We have made the following changes to further detail the phenotypic differences between the two disorders:

1. After obtaining detailed phenotypic information for additional patients and including them in the analysis, we now find that abnormality of the cerebellum is strongly enriched in the recessive cohort (OR = 22, FDR corrected $P = 4.1 \times 10^{-4}$).
2. We have added **Supplementary Table 4** which is dedicated to the MRI findings, broken down by individual features. **Supplementary Table 6** also includes the same data from nine individuals with ReNU syndrome, for comparison. We include the following text in the results section to describe this comparison: “To confirm these reported MRI differences, the same pediatric neuroradiologist who reviewed the images from individuals with the recessive NDD also reviewed MRI images for nine individuals with ReNU syndrome (**Sup. Table 6**). While 3/9 individuals with ReNU syndrome had mildly

dilated perivascular spaces in periventricular region, none had the severe dilation mimicking a confluent microcystic appearance that appears characteristic of the recessive disorder. Further, none of the nine ReNU individuals had cerebellar atrophy. Thinning of the corpus callosum was a common feature in both ReNU syndrome and the recessive NDD.”

3. We have conducted facial image analysis using GestaltMatcher, comparing 90 individuals with ReNU syndrome to 11 individuals with biallelic variants in *RNU4-2*. The results of this analysis are in **Figure 4B** and described through the following added text: “To further investigate the distinction in facial features, we utilised the GestaltMatcher framework¹⁶ to compare facial photographs for 90 individuals with ReNU syndrome, 11 individuals with biallelic variants in *RNU4-2*, and 100 ‘random’ individuals with different disorders. We calculated the distance between each pair of faces in clinical face phenotype space (CFPS; see **Methods**). We observed that pairs of individuals both with ReNU syndrome or both with biallelic *RNU4-2* variants were significantly closer (or more similar) than pairs of random individuals (linear mixed model $P=6.4 \times 10^{-54}$ and $P=1.8 \times 10^{-5}$ for ReNU and biallelic, respectively; **Fig. 4B**). In contrast, pairs where one individual was ReNU and the other biallelic *RNU4-2* were no closer in CFPS than random pairs ($P=0.37$). These data indicate there is convergence within, but distinction between, the facial phenotypes of the dominant and recessive disorders.”

Minor points

1. There are multiple affected siblings reported in the manuscript. Can the authors comment on phenotype concordance or divergent phenotypes among sibs? Please clarify in Supplementary Table 4 which individuals are siblings. A column with a family identifier could help.

Response: We have now added a column with a family identifier to both Supplementary Tables 1 and 3 (previously 4), and included a comment on phenotype concordance among siblings. This reads: “Phenotypic concordance was high across the seven multiplex families with detailed clinical data available (14 affected individuals). Siblings consistently shared a similar phenotype, including age at onset, systems affected and a comparable degree of intellectual disability (**Sup. Table 3**). In three families (F2, F4, F5), we observed variable expressivity: in each sibship, one sibling achieved independent walking whereas the other remained non ambulant. In Family F2, Individual 2 had more marked dilatation of perivascular spaces than his sibling, developed spasticity, and experienced seizure onset approximately ten years later.”

2. Although I would like to trust the aggregate genetic investigation of the collective team, the manuscript would benefit from a supplementary figure which includes pedigrees from all 32 individuals to show readers that due diligence was taken to genotype parents and unaffected siblings. If genotype data are missing from family members, this would be an opportunity to show this on the individual level.

Response: We have now added a new **Supplementary Figure 1** which includes pedigrees for all 28 families (38 individuals) in our characterised cohort.

3. Supplementary Table 1. Clarify the significance of the color-coded cells in the title. Do the different colors indicate recurrent variants?

Response: Yes you are correct. We have added the following to the legend for **Supplementary Table 1** to describe the reason for this colouring: “Coloured cells in the HGVS column highlight occurrences of the same variant.”

4. Please include the human genome reference build in Supplementary Tables 1, 2, 3.

Response: This has now been added to the column headers in each table.

5. Supplementary Table 4. Please avoid blank cells, and especially for phenotypic features, use NA (not available) or N (not present) to complete the entire table.

Response: We have replaced all blank cells in **Supplementary Table 3** (previously 4) with either N (not present) or NA (not available), as suggested.

6. Figure 1. SGE scores for insertions and deletions without available SGE data were inferred from the mean SGE score across all SNVs within the deleted nucleotides or taking the mean SGE score across all SNVs. Is there a published precedent for this, or are there experimental data that indicate that such an inference approach will not introduce false positive results?

Response: We have now included an analysis to justify our chosen approach in the methods section, including a new Supplementary Figure. This reads: “For insertions and deletions without available SGE data the function score was inferred from the mean SGE score across all SNVs within the deleted nucleotides or taking the mean SGE score across all SNVs within the nucleotides directly flanking the insertion. To validate this approach, we compared the SGE function score of 70 tested single base insertions and 8 tested single base deletions to the score that would be inferred using this approach. The experimental and inferred scores were strongly correlated (Spearman rank correlation coefficient = 0.83, $P < 0.001$ for insertions, and 0.83, $P = 0.015$ for deletions; **Sup. Fig 3A**). For most of the insertions and deletions that differed in their inferred and experimental scores by > 0.2 (22/33; 66.7%) the inferred score underestimated the deleteriousness of the variant (**Sup. Fig. 3B**), consistent with indels having a more severe effect than SNVs. Only four of the 78 variants (5.1%) had inferred scores that would cross the threshold of significance (< -0.302) but had a measured score below that threshold.” In addition, all of the variants with inferred scores that are included in our cohort have additional evidence in support of being pathogenic. For example, for three out of four (n.6_7insA, n.7_8insA, and 127_128del), other variants at overlapping positions are also found within our cohort. Further, all four variants are confidently within our SGE-identified important regions and are very rare in population cohorts UK Biobank and All of Us.

G. References: Citations are appropriate and current.

H. Clarity and context: The organization and writing style of the manuscript is clear.

Reviewer #3

In this manuscript, authors report a cohort of individuals with a neurodevelopmental disorder associated with biallelic variants in *RNU4-2*. De novo dominant-acting variants in the same gene have recently been described as causing the prevalent neurodevelopmental disorder ReNU syndrome. The authors demonstrate that biallelic variants associated with NDD are found outside of the ReNU syndrome region of the gene but clustered within regions which encode functionally important elements of U4. Biallelic variants in *RNU4-2* are rare in healthy population cohorts and enriched in cohorts with neurodevelopmental disorders. A companion manuscript performs saturation genome editing of *RNU4-2*, and individuals with NDD tend to have variants with a lower SGE score than individuals in the UKBB with biallelic *RNU4-2* variants. This is an important study that has been carefully conducted. Statistics are robust and appropriate. It is likely to be of interest to a broad readership including the neurology and genetics clinical communities, as well as those interested in the spliceosome and RNA biology. From a clinical perspective, awareness of both dominant and recessive conditions caused by variants in the same gene is crucial, in order to appropriately counsel families about the chance of future children in the family be affected.

Response: Thank you for your kind comments on our work.

Major comments:

1. Impact on splicing

Given the reduced sensitivity of the SGE assay for the biallelic variants, and the high rate of mutation rate in these genes, additional mechanistic information on the impact of these variants on splicing will be invaluable and will be of immediate value in clinical practice. Chen et al. previously characterised the impact on splicing by performing RNAseq on ReNU patient samples. A similar experiment should be performed on cell lines or blood samples from affected individuals with the recessive *RNU4-2* NDD. A common clinical scenario in the future will be interpreting *RNU4-2* variant/s in patients with a compatible but nonspecific phenotype, such as global developmental delay or intellectual disability, and as the SGE scores lack sensitivity for the recessive condition, such data will be invaluable. If access to patient-derived samples is not possible, the authors could use CRISPR/Cas9 to generate cellular models of the dominant and recessive conditions for splicing analysis.

Response: We strongly agree on the importance of such an analysis, however, at initial submission we did not have access to RNA-seq data from patients with the recessive NDD. We now have RNA-Seq data for six different patients, sequenced in two different contexts (three from cultured lymphoblastoid cell lines, three from RNA-Seq in whole blood). We use these data to demonstrate: (1) that individuals with the recessive *RNU4-2* NDD do not have the “ReNU signature” described in Nava *et al.* characterised by differences in 5' splice site usage, and (2) that individuals with the recessive NDD (but not those with dominant ReNU) have a reduction in *RNU4-2* transcript levels, consistent with a loss-of-function mechanism. This is consistent with what is known for variants in *RNU4ATAC* in the equivalent functional regions. Unlike in *RNU4ATAC* disorders, however, we do not see elevated intron retention in these recessive individuals. This may be due to a much subtler global effect, which would be expected, and is consistent with similar recent findings in *RNU2-2* (Jackson *et al.* medRxiv 2025; doi:10.1101/2025.09.02.25334957). The reduction in *RNU4-2* levels and associated dramatic shift in *RNU4-2/RNU4-1* ratio may be a highly specific biomarker for the recessive *RNU4-2* disorder. We now mention this in our discussion with the following text: “Using

RNA-sequencing data, we show that individuals with biallelic variants in *RNU4-2* have dramatically reduced levels of *RNU4-2* RNA, consistent with a loss-of-function mechanism. This is accompanied by an elevation in *RNU4-1* levels, consistent with potential compensatory upregulation. The *RNU4-2* to *RNU4-1* ratio discriminates strongly between cases and controls, and could therefore be a useful diagnostic biomarker for the recessive disorder.”

We have included these new results in a separate section at the end of the results (lines 539-584) and in a new Figure (**Fig. 6**). We have also added text to the abstract (lines 185-187), methods (lines 846-889), and discussion (lines 629-648) sections around these new findings.

2. Neuroradiological phenotypes

The phenotypes of individuals affected by dominant and recessive conditions seem strikingly similar, with the presence of periventricular perivascular dilation and cerebellar atrophy important distinguishing features of the recessive condition. Were MRI scans of individuals with ReNU syndrome specifically reviewed to look for signs of perivascular dilation and/or cerebellar atrophy? How does the perivascular dilatation reported here differ from the relatively common finding of cystic periventricular leukomalacia? Is this easy or difficult to distinguish? I note that the initial MRI reports for several individuals (presented in supplementary table 4) describe these anomalies as cystic periventricular leukomalacia. I feel this is important because a finding of cystic periventricular leukomalacia may be interpreted by clinicians as indicating an antenatal or perinatal insult (commonly seen in prematurity for example) and may reduce the likelihood of genetic testing being offered, particularly if the birth history is compatible. Secondly, some more detail on the phenotype of the cerebellar atrophy would be welcome – is this generalised, or affecting a particular region of the cerebellum?

Response: There is indeed some phenotypic overlap between the dominant and recessive conditions, and they are also similar to other syndromic NDDs. Hence, we agree that the MRI data are particularly important to distinguish the conditions. We have now added **Supplementary Table 4** which specifically displays the MRI features identified in 27 individuals from our recessive cohort and **Supplementary Table 6** which includes nine individuals with ReNU syndrome with MRI images assessed by the same experienced pediatric neuroradiologist. We have added the following to the main text to describe our review of these data: “To confirm these reported MRI differences, the same pediatric neuroradiologist who reviewed the images from individuals with the recessive NDD also reviewed MRI images for nine individuals with ReNU syndrome (**Sup. Table 6**). While 3/9 individuals with ReNU syndrome had mildly dilated perivascular spaces in periventricular region, none had the severe dilation mimicking a confluent microcystic appearance that appears characteristic of the recessive disorder. Further, none of the nine ReNU individuals had cerebellar atrophy. Thinning of the corpus callosum was a common feature in both ReNU syndrome and the recessive NDD.”

The dilatation of the perivascular spaces in the periventricular regions in this disorder is quite distinct from cystic periventricular leukomalacia. Cystic periventricular leukomalacia is most commonly seen in premature infants. The areas of white matter insult may undergo necrotic/cystic change over the course of days to weeks within the deep white matter, may

become confluent as larger cystic spaces, but almost invariably these cystic areas coalesce, decrease, and disappear within the first few months of life, leaving a low volume of residual white matter with configuration changes to the shape of the lateral ventricles (undulated border) and variable clinical features but often various degrees of cerebral palsy, among others. On the other hand, dilated perivascular spaces are linear and sometimes clustered dilatations of the Virchow-Robin spaces through which the deep white matter small vessels course through, and in the recessive *RNU4-2* cases, they were often located in the immediate vicinity of the lateral ventricles, albeit with varying degrees of dilatation. The most severe instances showed cyst like dilatations in some of the areas, but the linear configuration was nevertheless maintained overall. They were present postinfancy and quite distinct from cystic encephalomalacia when reviewed by experienced pediatric neuroradiologists. Upon central review of the available images, it was determined that the findings are dilated perivascular spaces in the periventricular white matter.

The cerebellar atrophy was not particularly limited to a region of the cerebellum and both the cerebellar hemispheres and vermis could be affected. This clarification was added to the manuscript with the following additional text: “The cerebellar atrophy was not limited to a specific portion of the cerebellum and both the cerebellar hemispheres and vermis could be involved.”

1. UKBB phenotypes

Can you comment on the phenotypes of the individuals in UKBB who are compound heterozygous for *RNU4-2* variants? What about the individual who has variants with significant SGE scores? Do you think that this indicates variable expressivity or penetrance?

Response: We have now investigated the phenotypes of the individuals in UK Biobank in more detail and include the following in the results section of the manuscript: “None of the eleven individuals had any evidence of neurodevelopmental or severe neurological phenotypes. All nine with information on the age at which they left education attended up to at least age 15. Four had a degree (4/11; 36.4%, versus 47.7% across the full cohort), 10/11 were reported as 'able to work' (90.9%, versus 93.4% across the full cohort) and none were outliers for fluid intelligence scores.”

The individual with the variant with the significant SGE score (n.120T>C) has compound heterozygous variants where the variant on the other allele (n.112A>C) has a non-significant score. Further, the n.112A>C variant is in the 3' stem loop region where we do not see any variants as homozygous or compound heterozygous in individuals with NDD. Hence, we believe that this individual is a carrier of a single pathogenic allele in the heterozygous state that is insufficient to cause the recessive phenotype.

2. SGE scores

The companion manuscript designates an SGE score <-0.39 as significant. However, this manuscript describes an SGE score of -0.2 as being the threshold that best distinguishes variants carried by individuals with a neurodevelopmental disorder from UKBB. However, both thresholds are used in this publication, which introduces confusion as to which is most appropriate to use clinically. I appreciate that the authors are reluctant to be too prescriptive, but a bit more clarity would be welcome – do they recommend the use of an average SGE score, or a threshold of -0.2 or -0.39 in evaluating biallelic variants, do they recommend that

this score differs based on the region that the variants lie within U4 (i.e., for putative ReNU syndrome de novo variants vs biallelic variants)?

Response: Thank you for pointing out that our use of different thresholds was confusing. We do not intend the mean SGE score of -0.2 to be interpreted as a definitive threshold. Rather, it was used to ensure confidence in the cases included in our initial characterisation of the disorder. We have edited the text where we first introduce the use of this mean score to make this clearer. It now reads: “We excluded five NDD individuals with mean SGE scores similar to those observed in the general population from further characterisation, using a threshold of -0.15 that maximally separated individuals in the UK Biobank from those with NDD (Fig. 1A). This threshold provides a distinction for analysis but should be interpreted as a pragmatic case definition for this initial characterisation rather than a definitive threshold (see Discussion).” Further, we now explicitly state that this threshold should not be used clinically in the discussion: “Here, we used the distribution of SGE scores of biallelic variants in the UK Biobank to determine a threshold for inclusion, defining a set of individuals with NDD falling outside this range for initial clinical characterisation. However, we do not recommend the use of this threshold in clinical settings without further calibration in independent cohorts.”

Of note, as part of our revisions to the companion manuscript, we have now performed a third replicate of the entire SGE experiment which has resulted in a slight shift to the SGE scores for all variants and recalculation of the thresholds. With this, an SGE function score of < -0.302 is now significant (rather than < -0.39) and we now use a mean SGE score < -0.15 across both alleles in an individual to include/exclude cases from our discovery cohort (rather than < -0.2).

Minor comments:

1. Table 1: Are the cerebellar anomalies being discussed cerebellar atrophy? In which case it would be useful to name this directly. Alternatively, if there are other anomalies observed, please give details.

Response: In response to a comment from reviewer #2, we have now added Supplementary Table 4 which is dedicated to the MRI findings, presented broken down by individual features with more consistent naming. This includes a column noting the cerebellar anomalies which are predominantly cerebellar atrophy.

2. Figure 5B:

a. A significant SGE score is given as function score < -0.39 , but earlier (in the comparison to UKBB variants) a threshold of -0.2 was applied. Would it not be more appropriate to use -0.2?

Response: In response to your below comments, we have now recoloured the variants across panels B (NDD) and D (UK Biobank) to all match the colouring in the SGE heatmap in panel C. This removes the need for two distinct categories, and makes the ranges of SGE scores for variants found in NDD cases and UK Biobank population controls much clearer. That said, we do still mark variants with significant SGE scores with an asterisk. Here we use the threshold of < -0.302 (previously -0.39) as this is the threshold of statistical

significance calculated directly from the SGE experimental data. As explained above, the mean score of <-0.15 (previously -0.2) is not intended as a definitive threshold.

b. It would be helpful to continue the shading colours from 5A to describe the functional domains. For example, could the ReNU syndrome critical regions be shaded in light blue/green, rather than darker gray?

Response: We have now ensured the shading colours are consistent throughout Figure 5, with the light teal marking the ReNU regions in panel A also used in the lower panels.

c. E it would be useful to plot the homozygous individuals in UKBB as solid circles together with the heterozygotes in D, or immediately below them, rather than with the UKBB allele frequency information, I found it hard to work out exactly where these variants are situated with respect to the SGE heatmap and functional domains (I think the position of the variant is indicated by a grey bar behind the UKBB allele frequency plot, but I'm not fully certain).

Response: We have now moved the homozygous points up to directly below the heatmap so that this is clearer and have plotted them as solid circles to be consistent with the representation for NDD individuals.

d. I presume in 5E colour reflects UKBB allele frequency? It would be useful to have the parameters of this in the key.

Response: We have changed the colouring of the heterozygous variants in UK Biobank (previously panel E) to reflect the SGE score. This is now consistent throughout the full plot. The height of the ellipse shows the greatest minor allele frequency at each site. This is now stated in the legend: "For heterozygous variants, the height of each ellipse is proportional to the logarithm of the allele count for the most frequent variant at that position (maximum allele count = 1,625) and the colour represents the SGE score of that variant, consistent with panel C."

e. It would be useful to colour the UKBB individuals' variants in D and E in the same way that the NDD individuals' variants are coloured in B, to make clear which are the variants that have a significant SGE score.

Response: We have recoloured the variants across panels B (NDD) and D (UK Biobank) to all match the colouring in the SGE heatmap in panel C so that this is consistent through the entire plot.

Supplementary Table 4

1. This is a useful table for a clinician to get a feeling for these patients and how they present, thank you for including this table and the detailed information.
2. However, it is quite hard to read the table in the current format because there are no row-lines, and information runs over several pages, often without headers, making it hard to work out which individual some of the features pertain to. My apologies if this is to do with how I opened the file. This would be much better as an excel spreadsheet. Alternatively, please orientate in a landscape direction, and provide the individual's number as the first column on each page, and the column headings on each page.

Response: We can assure you that the table is formatted as an Excel sheet, with clear row numbers. The PDF rendering on submission transformed the formatting. We hope that you can also see the 'original file' in the submission system if you would like to review this. Of note, this is now numbered as Supplementary Table 3.

3. Why does it start with individual 2, not 1, is this information missing, or is this a numbering error?

Response: We do not have detailed clinical information available for individual #1. We only include individuals in Sup Table 3 where we have detailed clinical information available. These numbers are consistent with the other tables and figures. We have now updated the legend to state: "Only individuals included in the discovery cohort and for whom we had detailed clinical information are listed here."

4. There are multiple spelling mistakes, a spell-check would be useful.

Response: We have tried our utmost to correct any spelling mistakes in the table.

5. I don't understand what 'false routes' in the other phenotypes section means: e.g. 'with false routes and apnoeas during crying' and 'stridor associated with false routes'. Does this mean trachea-oesophageal fistula? Does it mean aspiration of liquid into the lungs? Does it mean coughing/stridor after feeds/with secretions?

Response: Thank you for highlighting this lack of clarity. The term "false routes" was used to describe episodes of swallowing dysfunction with aspiration. It does not specifically imply a structural anomaly such as a tracheoesophageal fistula, but rather a functional swallowing disorder resulting in aspiration or choking during feeding or crying. We have replaced "false routes" with "oropharyngeal aspiration" throughout the text to make this clearer.

6. Please report anthropometric measurements such as birth weight and heights in a standard way, e.g. the measure plus the gestational age corrected centile. For some measurements no centiles are given, for others Z-scores are given, for others a notation such as [3.P] is given, which I'm afraid I don't understand the significance of. Consistency would be useful, I would recommend centiles.

Response: We have standardised all anthropometric data to sex specific and gestational age specific INTERGROWTH 21st standards, presenting measurements when available as raw values with centiles rather than the previous mix of centiles and z scores.

7. Consistency of reporting of EEG features would be useful. e.g. for each EEG comment on:

- a. is the background normal/abnormal?
- b. Were spike-wave discharges seen; if so, are they focal or generalised; if focal where?
- c. Were epileptic seizures seen; were they focal or generalised; if focal, where?
- d. Any other specific features suggestive of a particular epilepsy syndrome.

One entry refers to 'epilepsy predisposition', what does this mean (ie it would be more useful to give the observations that were made on the EEG that led the neurophysiologist to this conclusion)?

Response: We appreciate the helpful suggestion to ensure consistent and systematic reporting of EEG features. In response, we have now revised Supplementary Table 3 to include a structured, itemized set of fields to report EEG observations that align directly with the reviewer's recommended framework.

For every EEG report we now record:

- Whether the background was normal or abnormal
- Presence or absence of spike wave discharges
- Classification of spike wave discharges as focal or generalized and specific location when focal
- Presence or absence of electrographic seizures
- Classification of seizures as focal or generalized and specific location when focal
- Any additional EEG features that could suggest a particular epilepsy syndrome

Regarding the term "epilepsy predisposition", we agree that this wording was ambiguous. The original entry referred to an impression noted on a sleep deprived EEG while a 24 hour study was still pending. With the updated information now available, this individual's EEG findings are reported objectively as focal temporal activity with recorded epileptic seizures. The specific observations are now reported within the standardized EEG categories.

8. It would be useful, if possible, to know whether epilepsy was controlled with medication or was drug resistant/required multiple anti-epileptic medications.

Response: We have added a dedicated field in Supplementary Table 3 to report antiseizure medications used and pharmacoresistance. In the revised manuscript results, we now report: "Seizures occurred in 19/31 individuals (63.3%) with a median onset of 2.2 years. Seizure types at onset varied and included tonic clonic, atonic, focal, generalized, febrile, absence, and startle-triggered seizures. In most cases, seizure semiology evolved over time. Among the 19 individuals with seizures, 17 received antiseizure treatment. Most were treatment responsive: 13 received monotherapy and 4 required more than one antiseizure medication. Three individuals (15.8%) had pharmacoresistant epilepsy with persistent daily seizures. No individuals were reported to have experienced status epilepticus (**Sup. Table 3**)."

We have responded to the remaining minor points from the reviewers as detailed below.

Reviewer #2

1. Line 354, Results: The authors state that phenotype concordance was high across the seven multiplex families, however variable expressivity was observed in 3 families. The latter argues against high concordance, so the statement of “high concordance” should be softened.

This has been re-phrased to: “There was phenotypic concordance across the seven multiplex families with detailed clinical data available (14 affected individuals).”

2. Line 454, Results: Correct the p-values to include superscript.

This has been corrected.

Reviewer #3

3. SGE scores

The clarifications provided by the authors are helpful, thank you. I presume that the authors’ suggest that clinical labs should apply an evidence strength of supporting when variants have statistically significant SGE scores (as they themselves have done when classifying variants according to the ACMG guidelines), until an independently ascertained cohort is available to properly calibrate the scores? It would be helpful to add a sentence to this effect to the relevant section of the discussion.

The following text has been added to the discussion: “Full calibration of the SGE scores for use in variant classification for the recessive NDD will need to be performed in independent cohorts. In the meantime, we suggest that an evidence strength of supporting should be used for variants that have statistically significant SGE scores.”